# Game-Theoretic LLM: Agent Workflow for Negotiation Games

## Abstract

We investigate the rationality of large language models (LLMs) for strategic decision making via a game-theoretic perspective. We evaluate several state-of-the-art LLMs across a spectrum of complete-information and incomplete-information games. Our findings reveal that LLMs frequently deviate from rational strategies, particularly as the complexity of the game increases with larger payoff matrices or deeper sequential trees.

To address these limitations, we design multiple game-theoretic workflows that guide the reasoning and decision-making processes of LLMs. These workflows aim to enhance the models' ability to compute Nash Equilibria and make rational choices, even under incomplete information. Experimental results demonstrate that adopting these workflows significantly improves the rationality and robustness of LLMs in game-theoretic tasks. Specifically, with the workflow, LLMs exhibit marked improvements in identifying optimal strategies, achieving near-optimal allocations in negotiation scenarios, and reducing susceptibility to exploitation during negotiations. Furthermore, we explore the meta-strategic considerations of whether it is rational for agents to adopt such workflows, recognizing that the decision to use or forgo the workflow constitutes a game-theoretic issue in itself.

Our research contributes to a deeper understanding of LLMs' decision-making capabilities in strategic contexts and provides insights into enhancing their rationality through structured workflows. The findings have implications for the development of more robust and strategically sound AI agents capable of navigating complex interactive environments. Code and data supporting this study are available at https://anonymous.4open.science/r/game_theory-B7B5.

## 1 Introduction

Large Language Models (LLMs), such as GPT-4 and Claude, have achieved remarkable progress in natural language understanding and generation (Zhang & Ding, 2024; Ding et al., 2024; Fang et al., 2024), driving advancements in fields ranging from conversational AI (Dam et al., 2024; Dong et al., 2023) to content creation (Liang et al., 2024; Shao et al., 2024) and agentic task delegation (Guo et al., 2024b; Agashe et al., 2023; Xi et al., 2023). LLMs are increasingly integrated into applications that influence everyday activities, such as planning, acting, and decision-making. Therefore, the ability of LLMs to navigate complex situations has significant implications for their deployment in applications requiring strategic interaction, such as automated negotiations, economic modeling, and collaborative problem-solving (Bianchi et al., 2024; Horton, 2023; Li et al., 2024a; Chen et al., 2024; Li et al., 2023c).

Despite the wide exploration and utilization, LLM's capacity for rational behavior, particularly in strategic settings represented by game theory, remains an open question (Leng & Yuan, 2023; Stade et al., 2024; Wu et al., 2024b; de Zarzà et al., 2023; Lan et al., 2023). In this context, rationality implies an agent's ability to make decisions that maximize expected utility based on available information, an essential component of intelligent and adaptive decision-making. In the realm of game theory, rational agents are expected to act strategically, considering not only their own preferences but also the potential actions and preferences of

others. This is especially critical in incomplete-information games, where uncertainty about other players' information necessitates sophisticated reasoning and belief updating.

This paper investigates the capacity of LLMs to behave rationally in game-theoretic scenarios and explores methodologies to enhance their rational decision-making capabilities. We begin by assessing the performance of several state-of-the-art LLMs, including Claude-3.5 Sonnet, Claude-3 Opus, GPT-4o and o1 (Zhong et al., 2024), in both complete-information and incomplete-information games such as the Prisoner's Dilemma, Battle of the Sexes, the Escalation Game, and Deal-or-No-Deal (Lewis et al., 2017), presented in Figure 1. Our analysis reveals LLMs often deviate from rational strategies, particularly as the complexity of the game increases with larger payoff matrices or deeper sequential trees (Section 4). More experiments on their robustness against noise and uncertainty are presented in Appendix A.3.

**To address these limitations, we introduce a novel approach by proposing game-theory-inspired workflows specifically designed to guide the reasoning and decision-making processes of LLMs. This is the first attempt to systematically integrate classic game-theoretic strategies into LLM-based agent workflow, aiming to enhance their rational behavior and decision-making capabilities in strategic settings.** These workflows incorporate principles such as *Dominant Strategy Search*, which involves identifying strategies that yield the highest payoff regardless of the opponent's actions; *Backward Induction*, a method of solving extensive-form games by analyzing them from the end states backward to the initial decision nodes to determine optimal strategies; and *Bayesian belief updating*, which allows agents to refine their beliefs about other players' valuations based on observed actions and signals during the game. Cringed on these well-defined and well-studied game-theoretic methods, we design algorithms to guide the behavior and thinking process of LLM-based agents.

Additionally, we integrate fairness considerations like envy freeness and pareto optimality, which promote equitable and efficient outcomes in negotiations by ensuring that no agent prefers another agent's allocation to their own and that no improvements can be made without making at least one agent worse off.

---

**Contribution Summary**

- Comprehensive Evaluation of LLMs in Strategic Games and Identification of Rationality Limitations in LLMs (Section 4): Through empirical analysis, we uncover that LLMs often fail to behave rationally in strategic settings, exhibiting a lack of robustness to noise and randomness.
- Design of Game-Theory-Inspired Workflows (Section 4.4 and 5.2): We develop novel workflows inspired by game-theoretic concepts to guide the reasoning and decision-making processes of LLMs, incorporating analysis and algorithms from classic game theory.
- Emerging Research Direction (Section 5.5.3 and 5.6): Through the application of workflows, we identify a promising new research direction in meta-strategy, specifically focusing on the decision of whether to adopt a workflow and, potentially, which workflow to employ in varying scenarios.

---

## 2 Related Work

**LLMs in game-theoretic environments**  Understanding strategic behaviors (Mei et al., 2024) of LLMs entails important societal ramifications, as online users increasingly rely on intelligent assistants to interact with other agents, potentially also LLMs. To characterize LLM's behaviors, prior studies (Gemp et al., 2024; Sreedhar & Chilton, 2024; de Zarzà et al., 2023; Wu et al., 2024b; Akata et al., 2023; Fan et al., 2024; Mao et al., 2023; Guo, 2023) often adopt game theory, a mathematical framework that models cooperative behaviors of humans. These analyses involve comparing qualitative behaviors of LLM interactions against stylized entities, such as pareto optimal solutions and subgame-perfect equilibria (Fudenberg & Tirole, 1991). LLM research in game-theoretic environments belongs to a growing body of work on multi-agent LLMs (Li et al., 2023b) and their evaluation (Huang et al., 2024; Duan et al., 2024). In particular, AvalonBench (Wang et al., 2023b) serves as a valuable platform for developing new multi-agent strategies. (Guo, 2023; Park et al., 2024) observe LLMs to perform in games driven by *self-interest*, but falter in those that require *coordination*, a behavior that emerges under altruistic and/or submissive personalities (Akata et al., 2023; Guo, 2023).

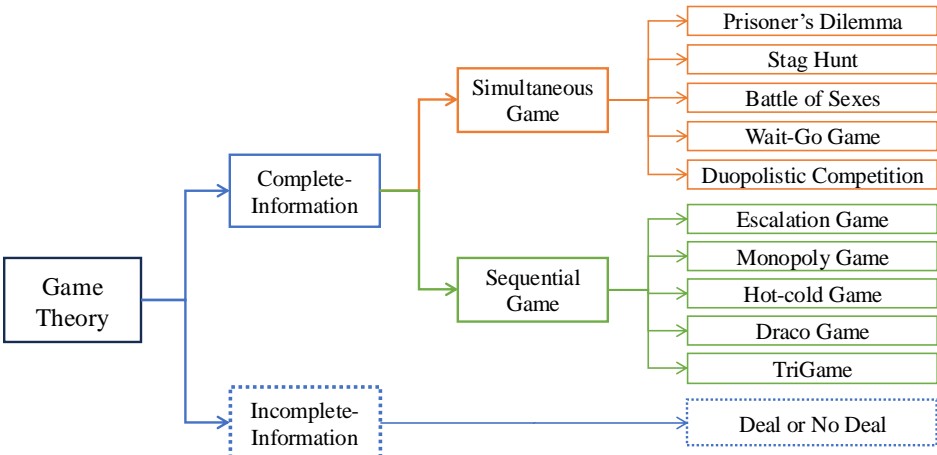

Figure 1: Game-theoretic Landscape Investigated in this Paper.

(Lorè & Heydari, 2023; Coda-Forno et al., 2024) observe different LLM families to exhibit varying levels of risk tolerance. Deferring details to experiments, we identify another source of brittleness as LLMs behave poorly when presented with numerically perturbed payoffs, even if they do not alter qualitative solutions to the game of interest. In short, there lacks a system that elicits *optimal* behaviors for LLMs in game-theoretic settings.

**Enhancing LLMs to solve games** (Gemp et al., 2024) is a representative approach that aims to elicit game-solving capabilities in LLMs. It models natural dialogues as incomplete-information games, and synthesizes optimal actions by instructing an LLM to respond with specific personalities. (Guo et al., 2024a) proposes an LLM-based self-play algorithm that emulates Monte-Carlo Tree Search to solve zero-sum games. Broadly, these methods belong to a family of prompting strategies (Wei et al., 2022; Yao et al., 2023; Liu et al., 2024) that tackle decision making instances. Our method is no exception, but differs as we imbue classical game theory in LLMs to allow for fine-grained control and analyses at each information state.

**Game-theoretic testbeds** Stylized games such as *Battle of Sexes*, *Prisoner's Dilemma*, *Rock-Paper-Scissors*, *Stag Hunt*, and *Ultimatum Game* have been extensively analyzed in the context of multi-agent systems (Sreedhar & Chilton, 2024; Akata et al., 2023; Fan et al., 2024; Lorè & Heydari, 2023). They represent a minimal setting characterized by small action spaces, limited number of terms, and the existence of analytical equilibria; yet they cannot capture complex interactions of real-world, multi-agent dialogues. Games that emulate real interactions are more challenging to analyze, but practically useful. Exemplar games in this category include *Deal-or-No-Deal* (Lewis et al., 2017), *Multi-Round Auction* (Mao et al., 2023), *Schedule-a-Meeting*, *Trading Fruit*, *Public Debate* (Gemp et al., 2024), *Avalon* (Wang et al., 2023b), *Pokémon* (Hu et al., 2024), *Chess* (Guo et al., 2024a) and *Bargaining* (Xia et al., 2024b). And there are efforts to collect multiple games and evaluate them (Duan et al., 2024). They often involve intractable action spaces (*e.g.* natural language) and do not attain analytical solutions; but they are nevertheless endowed with a well-defined payoff function for practitioners to analyze the optimality of their strategy, albeit in an end-to-end manner.

**Workflow-aided LLM-based agent** LLMs have been extensively utilized to decompose user requests and tasks, formulate plans, and employ various tools to execute actions (Ge et al., 2023; Wu et al., 2023; Li et al., 2023a; Light et al., 2023; Topsakal & Akinci, 2023; Xi et al., 2023). This reliance on the innate capabilities of LLMs has propelled advancements in domains such as natural language understanding, automated reasoning, and task automation. However, depending solely on the autonomous abilities of LLMs has revealed several notable limitations, including suboptimal performance (Xie et al., 2024; Xia et al., 2024a), lack of reliability

due to output randomness (Ingle et al., 2024; Li et al., 2024b; Schwartz et al., 2023), and the propagation of errors across sequential reasoning steps (Yu et al., 2023).

To address these challenges, the concept of incorporating agentic workflows (Wu et al., 2024a; Zeng et al., 2023; Xiao et al., 2024; Li et al., 2024c; 2023c) into LLM-based agents has emerged. Rather than allowing LLMs to independently decompose tasks and plan actions, workflows leverage human expertise and established knowledge frameworks to guide the direction and planning processes of the LLMs. Workflow-based agents have great potential to achieve high performance and adaptability in game scenarios (Xi et al., 2023). There have been multiple agents presented for real-world tasks (Jimenez et al., 2023; Yang et al., 2024), embodied agents that interact with the environment (Wang et al., 2023a; Zhu et al., 2023), agents that can learn from playing a text-based game (Xu et al., 2023), and agent operating on games with imperfect information (Guo et al., 2023).

**Game-theoretic Workflow in this Paper** In this paper, we integrate game-theoretic principles into the reasoning processes of LLMs prior to decision-making. By guiding the models to derive rational strategies and make decisions based on these strategies, we aim to enhance their ability to perform effectively in strategic settings.

To the best of our knowledge, this work is the first to combine LLM-based agents with structured workflows inspired by game theory to enhance strategic decision-making capabilities. We address both complete-information and incomplete-information settings, drawing upon classical game-theoretic principles to guide the reasoning processes of LLMs. Specifically, we introduce a novel algorithm for negotiation under incomplete information, enabling agents to perform Bayesian updates and make rational decisions based on limited knowledge of other players. This approach bridges the gap between traditional game theory and LLM-based agents, advancing the applicability of LLMs in complex strategic environments and opening new avenues for research at the intersection of AI and game theory.

## 3 Notations

In this section, we introduce the notations we will use throughout this paper. A game may involve $n$ agents, indexed by $i \in \mathbf{N} = \{1, 2, \ldots, n\}$. Each agent has an action set defined as: $A_i = \{a_i^1, \ldots, a_i^k\}$ where $k$ is the number of available actions for each player. The agents are also given the payoff matrix $U_i(\mathbf{a})$ for all strategy profiles $\mathbf{a} = (a_1, a_2, \ldots, a_n)$ for $a_i \in A_i$. If there are only two players, we use $i \in \{1, -1\}$ to denote the two players and the payoff matrix is $U_i(a_i, a_{-i})$ for all $a_i \in A_i$ and $a_{-i} \in A_{-i}$, where $A_{-i}$ is the action set of the other player. The payoff matrix $U$ is an $m \times n$ matrix where each entry $U_{ij}$ corresponds on the strategy profile $(a_1^i, a_{-1}^j)$ and contains the payoff vector $(u_1(a_1^i, a_{-1}^j), u_{-1}(a_1^i, a_{-1}^j))$.

Informally, the payoff matrix is typically structured as follows: (1) Rows represent the possible actions or strategies available to Player$_1$ (the *row player*). (2) Columns represent the possible actions or strategies available to Player$_{-1}$ (the *column player*). (3) Cells within the matrix contain ordered pairs of numbers $(u_1, u_{-1})$ where $u_1$ is the payoff to Player$_1$ and is the payoff to Player$_{-1}$ for the corresponding combination of strategies.

## 4 Complete-information Games

In this section, we delve into several classic game-theoretic scenarios involving complete information to assess whether LLMs can act as rational decision-makers in reaching Nash Equilibria and achieving Pareto optimal outcomes through negotiation, which is multi-agent multi-round conversation. We employ various game-theoretic settings to evaluate the rationality of LLM-based agents. Then we design game-theory-motivated workflows to guide and enable LLMs for better performance. We investigate the performance of workflow-aided LLMs and the impact of negotiation on performance.

### 4.1 Introduction to Complete-information Games TestBed

In our exploration of classic game-theoretic scenarios, we constructed a comprehensive testbed comprising 10 classic complete-information games to evaluate the rationality and strategic decision-making capabilities of LLM-based agents. This testbed includes 5 simultaneous-move games and 5 sequential-move games. For the simultaneous-move games, 3 are coordination games (Stag Hunt, Battle of Sexes, Wait-Go Games), wherein achieving the Pareto optimal Nash Equilibrium necessitates effective negotiation between agents. These games are instrumental in examining the agents' ability to communicate, build trust, and align their strategies for mutual benefit. Details of the games can be found in Appendix A.2.

### 4.2 Experiment Setting

To evaluate the negotiation performance of LLMs, we utilize four state-of-the-art LLMs as the backbone for agents in negotiation games: Claude-3.5 Sonnet (Sonnet), Claude-3 Opus (Opus), GPT-4o, and o1. For Claude-3.5 Sonnet, Claude-3 Opus, and GPT-4o, we set the temperature to 1.0 to encourage exploratory behavior in negotiation scenarios. For o1, we use the default temperature.

To assess the rationality of decision-making, we measure the ability of the agents to reach Nash Equilibrium. For games where there are multiple Nash Equilibria such as in the game Stag Hunt, we measure the ability of agents to reach the pareto optimal Nash Equilibrium. For each game scenario, we conduct 10 trials to mitigate the effects of randomness. In the tables, we report the percentage of cases in which Nash Equilibrium is achieved, providing a quantitative indicator of the agents' rationality across various negotiation contexts[1].

### 4.3 Evaluation on LLM's performance

In this evaluation, we assess the LLMs' performance using chain-of-thought prompting without negotiation (see Table 1 and with 4 rounds of negotiation (see Table 2).

| Games | Nash Equilibrium | | | | Pareto optimal Nash Equilibrium | | | |
|---|---|---|---|---|---|---|---|---|
| | Sonnet | GPT-4o | Opus | o1 | Sonnet | GPT-4o | Opus | o1 |
| Prisoner's Dilemma | 1.0 | 0.9 | 0.9 | 1.0 | - | - | - | - |
| Stag Hunt | 0.6 | 0.8 | 0.9 | 0.4 | 0.3 | 0.4 | 0.0 | 0.1 |
| Battle of Sexes | 0.3 | 0.2 | 0.6 | 0.5 | - | - | - | - |
| Wait-Go Game | 0.4 | 0.5 | 0.7 | 0.3 | - | - | - | - |
| Duopolistic competition | 0.3 | 0.2 | 0.1 | 0.7 | - | - | - | - |
| Escalation Game | 0.0 | 0.2 | 0.2 | 1.0 | - | - | - | - |
| Monopoly Game | 1.0 | 0.4 | 1.0 | 1.0 | - | - | - | - |
| Hot-cold Game | 0.9 | 0.1 | 0.7 | 1.0 | - | - | - | - |
| Draco Game | 0.3 | 0.0 | 0.7 | 0.9 | - | - | - | - |
| Trigame | 0.0 | 0.0 | 0.1 | 1.0 | - | - | - | - |
| average | 0.45 | 0.32 | 0.59 | **0.78** | - | - | - | - |

Table 1: Performance of LLM on complete-information games without negotiation

It is noteworthy that in most games, negotiation enhances the outcomes. In coordination games such as the Stag Hunt, Battle of the Sexes, and Wait-Go Game, negotiation is a fundamental component of decision-making, making the improvement in performance through negotiation an intuitive result. Additionally, in games like the Escalation Game, Draco Game, and Trigame, negotiation similarly enhances performance. This improvement can be attributed to the fact that during the negotiation phase, agents engage in more extensive computational deliberation during negotiation to determine the most suitable actions.

---

[1]Notice that the LLM-based agent's performance on these games can be largely affected by the numerical instantiations of the payoff matrix. Relevant research and observations are presented in Section A.3. Here in this experiment, we use the payoff matrices presented in the previous subsection.

| Games | Nash Equilibrium | | | | Pareto optimal Nash Equilibrium | | | |
|---|---|---|---|---|---|---|---|---|
| | Sonnet | GPT-4o | Opus | o1 | Sonnet | GPT-4o | Opus | o1 |
| Prisoner's Dilemma | 0.0 | 0.1 | 0.0 | 0.9 | - | - | - | - |
| Stag Hunt | 1.0 | 0.9 | 1.0 | 0.9 | 1.0 | 0.9 | 1.0 | 0.9 |
| Battle of Sexes | 0.7 | 0.9 | 0.7 | 0.8 | - | - | - | - |
| Wait-Go Game | 1.0 | 0.8 | 0.4 | 0.8 | - | - | - | - |
| Duopolistic competition | 0.1 | 0.1 | 0.0 | 0.5 | - | - | - | - |
| Escalation Game | 0.6 | 0.4 | 0.6 | 1.0 | - | - | - | - |
| Monopoly Game | 1.0 | 0.9 | 1.0 | 1.0 | - | - | - | - |
| Hot-cold Game | 0.7 | 0.2 | 0.7 | 0.4 | - | - | - | - |
| Draco Game | 1.0 | 0.8 | 1.0 | 1.0 | - | - | - | - |
| Trigame | 0.2 | 0.5 | 0.6 | 1.0 | - | - | - | - |
| average | 0.63 | 0.56 | 0.60 | **0.83** | - | - | - | - |

Table 2: Performance of LLM on complete-information games with 4 rounds of negotiation. Results highlighted in red indicate scores lower than the LLMs' performance without negotiation.

For instance, in the Escalation Game, all evaluated LLMs except for model o1 are unable to identify the Nash Equilibrium in the absence of negotiation. However, when engaged in 4 rounds of negotiation, the agents are provided with the opportunity to deliberate and exchange various proposals and thus achieve a much better performance.

*However, certain games, notably the Prisoner's Dilemma and Duopolistic Competition, exhibit a different dynamic where negotiation significantly diminishes performance.* **This situation stems from the fact that in these games, the Nash Equilibrium is not pareto optimal.** In the Prisoner's Dilemma, the pareto optimal strategy is for both players to cooperate, whereas the Nash Equilibrium occurs when both players defect. Following negotiation, all models except for o1 tend to adopt the pareto optimal strategy of mutual cooperation, which deviates from the Nash Equilibrium. This deviation arises from the pursuit of mutual benefit despite the lack of solid and guaranteed trust between players. While the performance for o1 remains high. Similarly, in Duopolistic Competition, the Nash Equilibrium does not align with the pareto optimal outcome. The pareto optimal strategy is for both players to choose Action 2, while the Nash Equilibrium is for both to choose Action 3. Empirical results demonstrate that without negotiation, Claude-3.5 Sonnet selects the pareto optimal strategy in 5 out of 20 total choices for both players, whereas after negotiation, this increases to 11 out of 20. For GPT-4o, the corresponding figures are 0 out of 20 without negotiation and 8 out of 20 with negotiation. Claude-3 Opus follows a similar pattern, increasing from 4 out of 20 choices without negotiation to 16 out of 20 after negotiation. While for o1 model, it only shows a marginal improvement from 4 out of 20 choices without negotiation to 5 out of 20 after negotiation.

These findings underscore that while negotiation generally enhances outcomes in coordination games and certain strategic scenarios, it can inadvertently undermine performance in games where the Nash Equilibrium is not pareto optimal. In such cases, negotiation may lead agents away from the rational strategy predicted by game theory. Notably, most LLMs – potentially all except for model o1 – appear to exhibit a vulnerability in their rational decision-making processes during negotiations. **When engaged in dialogue with another agent, these LLMs often place undue trust in the opponent's statements without sufficient justification.** The use of persuasive or amicable language by the opposing player can lead LLMs to make decisions that deviate from the rational strategies prescribed by game-theoretic analysis.

### 4.4 Workflow Design based on Classic Game Theory

In this section, we present the workflow employed for complete-information games, which leverages classic game-theoretic strategies to guide decision-making and optimize outcomes. This structured approach aims to align LLMs' responses with rational game-theoretic principles, thereby enhancing their ability to identify optimal strategies and maintain robust rationality, even in the context of negotiation. Through this workflow,

we assess whether LLMs can sustain rational choices and avoid strategic vulnerabilities, particularly in scenarios where negotiation might otherwise lead to suboptimal or exploitable decisions, *i.e.* pareto optimal strategies that are not Nash Equilibrium.

### 4.4.1 Workflow for Simultaneous Game

In the simultaneous game workflow, each agent (player) seeks to determine their optimal strategy by considering both their own possible actions and the potential responses of the other player. This involves conditional reasoning, generating thinking chains, and summarizing these into an overall strategy under the guidance of the workflow. Here, we will explain the workflow with corresponding mathematical formulations.

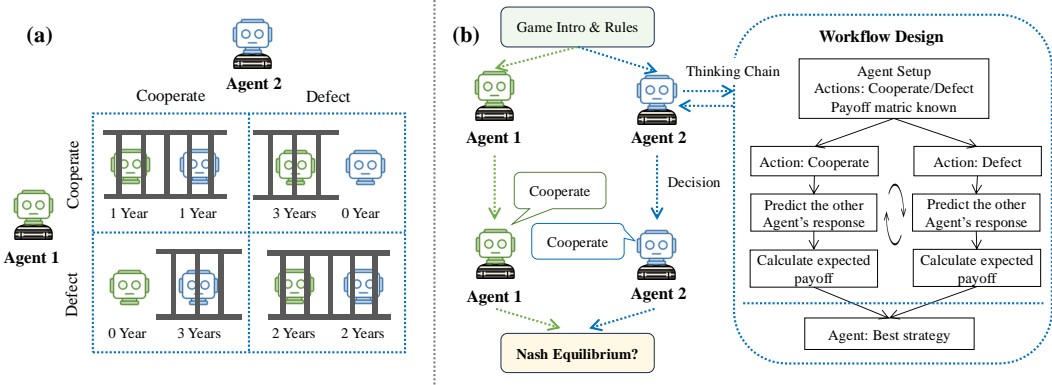

Figure 2: An illustration of workflow design for simultaneous game. (a) Illustration of prisoner's dilemma. (b) Workflow design for prisoner's dilemma.

**Game Setup**   Each agent $i \in \{1, -1\}$ begins by interpreting the game introduction and rules, which include detailed descriptions of the payoff matrices. The agents are provided with the exact payoffs for all possible action combinations. They are given the action sets

$$A_i = \{a_i^1, \ldots, a_i^k\}$$

where $k$ is the number of available actions for each player. The agents are also given the payoff matrix $U_i(a_i, a_{-i})$ for all $a_i \in A_i$ and $a_{-i} \in A_{-i}$, where $A_{-i}$ is the action set of the other player.

**Strategy Formulation**   With full knowledge of the payoff matrix, agents perform best response analysis to determine their optimal strategies. The goal for each player is to choose an action $a_i^*$ that maximizes their own payoff, anticipating the rational response of the other player. The optimization is done by iterating over the player's own actions and predicts the opponent's responses. It also considers the opponent's possible actions and determines their best responses:

For each possible action $a_i$ that player $i$ can take: the LLM-based agent computes the opponent's best response based on the resulting payoff by computing $a_{-i}^*(a_i) = \arg\max_{a_{-i} \in A_{-i}} U_{-i}(a_i, a_{-i})$ and the corresponding expected payoff $U_i(a_i, a_{-i}^*(a_i))$. This basically means that if player $i$ chooses $a_i$, then the other player will choose $a_i^*(a_i)$ resulting in a payoff of $U_i(a_i, a_{-i}^*(a_i))$. In the workflow, the LLM is guided to compute $a_{-i}^*(a_i)$ and $U_i(a_i, a_{-i}^*(a_i))$.

In addition, for each possible action $a_{-i}$ that the opponent might take, the LLM-based agent is guided to compute their best response as well: $a_i^*(a_{-i}) = \arg\max_{a_i \in A_i} U_i(a_i, a_{-i})$ and the corresponding expected payoff $U_i(a_i^*(a_{-i}), a_{-i})$. This basically means that if the opponent chooses $a_{-i}$, then $i$ will choose $a_i^*(a_{-i})$ resulting in a payoff of $U_i(a_i^*(a_{-i}), a_{-i})$.

After compiling all thinking chains, the agent is guided to summarizes them to into a comprehensive set of strategic considerations, aiming to find an action profile $(a_i^*, a_{-i}^*)$ that constitutes a Nash Equilibrium such

that

$$\forall a_i \in A_i, U_i(a_i^*, a_{-i}^*) \geq U_i(a_i, a_{-i}^*)$$
$$\forall a_{-i} \in A_{-i}, U_{-i}(a_i^*, a_{-i}^*) \geq U_i(a_i^*, a_{-i})$$

By considering the best responses and counter-responses, agents implicitly search for a Nash Equilibrium through their strategic reasoning. Figure 2 presents the workflow in diagram.

### 4.4.2 Workflow for Sequential Game

For sequential game, we adopt the traditional game-theoretic method: backward induction. Backward induction is a fundamental method in game theory used to solve sequential games with complete information. It involves analyzing the game from the end backward to the beginning, determining the optimal strategy at each decision point by considering the future consequences of current actions.

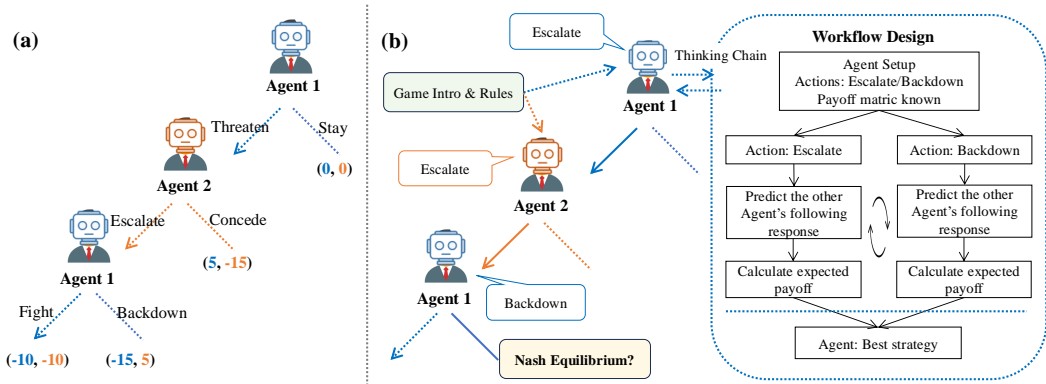

Figure 3: An illustration of workflow design for sequential game. (a) Illustration of escalation game. (b) Workflow design for escalation game.

**Game Setup**  Each agent $i \in \{1, -1\}$ begins by interpreting the game introduction and rules, which include detailed descriptions of the payoff matrices. In a sequential game, players make decisions one after another and for each action, the player is fully aware of all previous actions taken. The game can be represented as a game tree with nodes being decision points and edges being actions.

**Notations**

- $H$ be the set of non-terminal decision nodes in the game tree.

- $Z$ be the set of terminal nodes (endpoints of the game).

- $A(h)$ be the set of possible actions available at node $h \in H$

- player$(h)$ be the player who makes a decision at node $h$

- $U_i(z)$ be the payoff to player $i$ at terminal node $z \in Z$

We define the value function $V_i(h)$ for player $i$ at node $h$ as the maximum expected utility the player can achieve from that node onward. For terminal nodes $z \in Z$:

$$V_i(z) = U_i(z)$$

For decision nodes $h \in H$, if it is player $i$ who moves at node $h$:

$$V_i(h) = \max_{a \in A(h)} V_i(h \cdot a)$$

if it is the other player $j$ moves at node $h$:

$$V_i(h) = V_i(h \cdot a^*(h))$$

where $h \cdot a$ denotes the successor node reached from $h$ after action $a$ and $a^*(h)$ is the optimal action chosen by player $j$ at node $h$:

$$a^*(h) = \arg\max_{a \in A(h)} V_j(h \cdot a)$$

**Strategy Formulation** The backward induction starts from terminal nodes. For each terminal node $z \in Z$, set $V_i(z) = U_i(z)$ for all player $i$, and then proceed to preceding decision nodes. Therefore, for each decision node $h$, the LLM-based agent is guided to comput the optimal action $a^*(h)$ and value $V_i(h)$ based on the player who moves at $h$.

To determine the optimal action, if player$(h) = i$, which is the player in question, LLM is guided to compute:

$$a * (h) = \arg\max_{a \in A(h)} V_i(h \cdot a)$$

Otherwise, if player$(h) = j \neq i$, assuming player $j$ will choose their optimal action $a^*(h)$, LLM is guided to compute:

$$a^*(h) = \arg\max_{a \in A(h)} V_j(h \cdot a)$$

Then, $V_i(h) = V_i(h \cdot a^*(h))$. The workflow directs the LLM to employ backward induction by systematically traversing the game decision tree from the terminal nodes back to the initial node to formulate an optimal strategy. The strategy derived from this backward induction process is subsequently incorporated into the contextual framework for each decision-making and negotiation step, thereby guiding the LLM's strategic choices throughout the game. Figure 3 presents the workflow in diagram.

### 4.5 Experiments for Classic Game Theory with Workflow

In this section, we present the results of LLM-based agents employing the workflow to summarize strategies in Table 3 and 4. We begin by examining the experimental outcomes without negotiation. Notably, with the introduction of the workflow, the performance of all language models, except for o1, has improved significantly.

| Games | Nash Equilibrium | | | | Pareto optimal Nash Equilibrium | | | |
|---|---|---|---|---|---|---|---|---|
| | Sonnet | GPT-4o | Opus | o1 | Sonnet | GPT-4o | Opus | o1 |
| Prisoner's Dilemma | 1.0 | 1.0 | 1.0 | 1.0 | - | - | - | - |
| Stag Hunt | 1.0 | 1.0 | 1.0 | 1.0 | 1.0 | 1.0 | 1.0 | 1.0 |
| Battle of Sexes | 0.3 | 0.3 | 0.4 | 0.5 | - | - | - | - |
| Wait-Go Game | 0.2 | 0.6 | 0.1 | 0.3 | - | - | - | - |
| Duopolistic competition | 1.0 | 0.5 | 0.6 | 0.3 | - | - | - | - |
| Escalation Game | 0.5 | 0.2 | 0.3 | 1.0 | - | - | - | - |
| Monopoly Game | 1.0 | 1.0 | 1.0 | 1.0 | - | - | - | - |
| Hot-cold Game | 1.0 | 1.0 | 1.0 | 1.0 | - | - | - | - |
| Draco Game | 1.0 | 1.0 | 1.0 | 1.0 | - | - | - | - |
| Trigame | 0.6 | 0.9 | 0.7 | 1.0 | - | - | - | - |
| average | 0.76 | 0.69 | 0.71 | **0.81** | - | - | - | - |

Table 3: Performance of workflow-LLM on complete-information games without negotiation

The Stag Hunt game achieves the pareto optimal Nash Equilibrium even without negotiation; that is, coordination is established when trust is ensured by the payoff matrix and the assumption of the other player's

rationality. In the Duopolistic Competition game, performance increases from near-zero to a substantially higher level. However, it is important to observe that the performance of model o1 declines considerably in this context. Analyzing this decrease is challenging because the intermediate chain-of-thought is not accessible, limiting our ability to diagnose the cause. Other games, such as the Monopoly Game, Hot-Cold Game, Draco Game, and Trigame, also exhibit substantial performance improvements. This indicates that the workflow effectively guides LLMs in computing the Nash Equilibrium.

*Despite these advancements, the performance in the Escalation Game remains low. Upon examining the reasoning process, we find that the LLMs appear somewhat insensitive to numerical values, particularly when the Nash Equilibrium results in a reward of 0 for both players.* The LLMs seem less inclined to select strategies leading to such outcomes. Below, we provide an example of strategy summarization generated by the Opus model.

> **Example Wrong Strategy Derived for Escalation Game**
>
> Based on the thinking chains, here is the strategy I would take asAlice in this sequential game:
>
> Always choose Action_2, regardless of what actions have been taken previously in the game by either player.
>
> The reasoning is:
> - If I choose Action_1 on my first turn, both players receive 0 reward.
> - If I choose Action_2 on my first turn, I have a chance at a positive reward of 1 (if Bob picks Action_1 next), and at worst I will receive -1 reward (if Bob picks Action_2 and I pick Action_2 again). A potential 1 or -1 reward is better than a guaranteed 0.
> - Whenever faced with a choice between Action_1 and Action_2 later in the game after initially picking Action_2, Action_2 always gives a better reward for me (-1 instead of -2).

With the implementation of four rounds of negotiation, we observe that there is no significant trend toward adopting pareto optimal strategy profiles in games where the Nash Equilibrium is not pareto optimal. Notably, for models such as Claude-3.5 Sonnet and GPT-4o, there is a slight decrease in performance in games like the Hot-Cold Game, Draco Game, and TriGame – games in which these models already exhibited suboptimal performance. This decline does not stem from the inherent properties of these games but rather from the influence of multi-round negotiations causing deviations from the strategies computed and guided by the workflow.

| Games | Nash Equilibrium | | | | Pareto optimal Nash Equilibrium | | | |
|---|---|---|---|---|---|---|---|---|
| | Sonnet | GPT-4o | Opus | o1 | Sonnet | GPT-4o | Opus | o1 |
| Prisoner's Dilemma | 1.0 | 0.9 | 0.9 | 1.0 | - | - | - | - |
| Stag Hunt | 1.0 | 1.0 | 1.0 | 1.0 | 1.0 | 1.0 | 1.0 | 1.0 |
| Battle of Sexes | 0.7 | 0.8 | 1.0 | 0.8 | - | - | - | - |
| Wait-Go Game | 0.9 | 0.7 | 0.6 | 0.7 | - | - | - | - |
| Duopolistic competition | 0.6 | 0.7 | 0.6 | 0.3 | - | - | - | - |
| Escalation Game | 0.3 | 0.2 | 0.4 | 1.0 | - | - | - | - |
| Monopoly Game | 1.0 | 1.0 | 1.0 | 1.0 | - | - | - | - |
| Hot-cold Game | 0.7 | 0.8 | 1.0 | 1.0 | - | - | - | - |
| Draco Game | 0.8 | 1.0 | 0.9 | 1.0 | - | - | - | - |
| Trigame | 0.3 | 0.8 | 0.9 | 1.0 | - | - | - | - |
| average | 0.73 | 0.78 | 0.83 | **0.88** | - | - | - | - |

Table 4: Performance of workflow-LLM on complete-information games with 4 rounds of negotiation

> **Intermediate Summary**
>
> - **LLM Performance Without Workflow**: All evaluated LLMs, except for model o1, demonstrate significantly poor performance in game-theoretic tasks when confronted with larger payoff matrices or deeper sequential decision trees. This indicates a limitation in their ability to handle complex strategic reasoning without additional structural guidance.
> - **Effect of Negotiation Without Workflow**: In the absence of the workflow, negotiation tends to systematically shift the strategies of LLMs away from the Nash Equilibrium toward non-equilibrium pareto optimal strategies, with the exception of model o1.
> - **LLM Performance With Workflow**: The implementation of the workflow markedly enhances the performance of LLMs in game-theoretic contexts. The agents become more adept at computing Nash Equilibria and selecting optimal strategies. However, certain challenges persist, such as in the Escalation Game, where numerical computation errors hinder deriving the Nash Equilibrium following the workflow. This underscores that while the workflow improves rational decision-making, it does not entirely eliminate all computational limitations.
> - **Impact of Negotiation With Workflow**: Even when utilizing the workflow, negotiation can still divert LLMs from adhering strictly to the Nash Equilibrium strategies derived through systematic reasoning. The agents may be influenced by negotiation dialogues to consider alternative strategies. Nevertheless, unlike the scenario without the workflow, negotiation does not systematically shift the agents toward pareto optimal strategies at the expense of Nash Equilibrium compliance.

## 5 Incomplete-information Game with Negotiation

Building upon our exploration of LLMs' capabilities in classical game-theoretic scenarios and the development of workflows to enhance their reasoning processes, we now shift our focus to a more realistic and complex setting: incomplete-information games. Specifically, we consider a common resource allocation scenario where a shared pool of resources must be distributed among several players. Each player possesses private valuations of the resources that sum to a common total value, reflecting their individual preferences and priorities. Crucially, no player has knowledge of the other players' valuations, embodying the incomplete-information aspect of the game.

Our workflow addresses this general class of incomplete-information games by enabling agents to reason under uncertainty and update their beliefs based on observed actions and communications during negotiation. We evaluate the performance of LLMs in this setting and propose a workflow designed to guide their decision-making processes effectively. Experiments are conducted using the standard representative game dataset "Deal or No Deal" (Lewis et al., 2017), which provides a suitable framework for testing and validating our approach in handling incomplete information during negotiations.

### 5.1 Introduction to Common Resource Allocation with Private Valuation

Here we provide a formal definition for the incomplete-information game that we will focus in this section.

**Definition 1** (Common Resource Allocation with Private Valuation)**.** *Consider a game involving a set of players* $\mathbf{N} = \{1, 2, \ldots, n\}$ *and a set of items or resources* $K = \{1, 2, \ldots, k\}$*. Each player* $i \in \mathbf{N}$ *possesses a private valuation vector* $\mathbf{v}_i = (v_i^1, v_i^2, \ldots, v_i^k)$ *where* $v_i^k \geq 0$ *represents the value that player* $i$ *assigns to item* $k$*. These valuations reflect the individual preferences of the players and are private information; that is, each player knows their own valuations but not those of the other players.*

*The valuations satisfy the normalization condition:*

$$\sum_{j=1}^{k} v_i^j = \mathcal{V} \quad \forall i \in \mathbf{N}$$

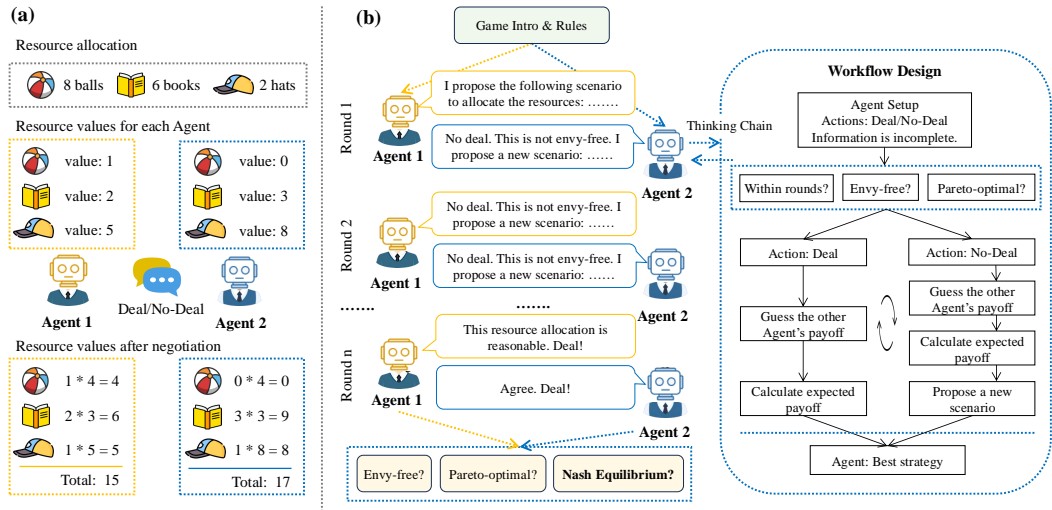

Figure 4: An illustration of workflow design for incomplete-information game with negotiation. (a) Illustration of deal/no-deal game. (b) Workflow design for deal/no-deal game.

*where $\mathcal{V}$ is a constant total value common to all players. This condition ensures that while players may value items differently, the total valuation of all items is the same for each player.*

*An allocation is a partition of the item set $K$ among the players, represented as $L = (L_1, L_2, \ldots, L_n)$ where $L_i \subseteq K$ is the set of items allocated to player $i$. The allocation must satisfy:*

$$L_i \cap L_j = \emptyset \quad \forall i \neq j \text{ and } \bigcup_{i=1}^{n} L_i = K$$

*Each player $i$ receives a utility $U_i$ based on the private valuations and the allocation:*

$$U_i = \sum_{k \in K} v_i^k \times \mathbb{1}(k \in L_i)$$

*where $\mathbb{1}(\cdot)$ is the characteristic function.*

The objective of each player is to maximize their own utility $u_i$ through negotiation with the other players. However, due to incomplete information where each player does not know the valuations of the other players, players must make decisions under uncertainty. Negotiation involves proposing and responding to allocation offers, during which players may communicate, share limited information, or infer the valuations of others based on their actions and statements. Players may also update their *belief systems* – probability distributions over the possible valuations of other players—using Bayesian inference as the negotiation progresses.

It is important to note that the Nash Equilibrium for such games, which are variants of the Ultimatum Game, occurs when the proposer offers the smallest possible amount and the responder accepts it. However, this outcome is rarely observed in real-life situations because fairness is often a significant concern (Nowak et al., 2000); unfair deals may be rejected even when accepting them is the rational choice in terms of utility maximization. Therefore, in this paper, we adopt a more realistic setting by designing a workflow that encourages reaching fair allocations while simultaneously maximizing self-interest as much as possible. Our workflow addresses a very general class of incomplete-information games, enabling agents to negotiate under uncertainty and strive for equitable outcomes that reflect both fairness and individual utility optimization.

Without loss of generality, we focus on the problem where there are only two players and we use $i \in \{1, -1\}$ to index the two players. Any rounds of negotiations are allowed.

## 5.2  Workflow Design

The workflows employed for complete-information games are based on well-established game-theoretic frameworks, leveraging the extensive research and thorough understanding available in this domain. **In contrast, common resource allocation problems characterized by incomplete information lack a standardized solution framework. To address this gap, we develop a novel algorithm for this setting.** This workflow is designed to guide the multi-round negotiation process, facilitating the attainment of allocations that are both mutually agreeable and optimized for each player's self-interest. Figure 4 presents a high-level diagram of the workflow.

**Assumption 1.** *Each participant to the game defined in Definition 1 is rational and attains the following objectives:*

- *Objective 1: Achieve an agreement (i.e., successfully complete the allocation).*

- *Objective 2: Maximize their own utility, given that an agreement can be reached.*

All negotiation proposals and communications between the players revolve around these two objectives. Regarding the first objective, we assume that an agreement can only be reached if the allocation satisfies the criterion of envy freeness. That is, each player must not prefer the allocation received by the other player over their own allocation. For the second objective, each player seeks to maximize their own utility under the condition that the agreement remains envy free. Essentially, players aim to maximize their rewards as much as possible while ensuring the allocation is envy free.

Following Definition 1, each agent's valuation of common resources remains private and undisclosed to others. Therefore, it becomes essential for agents to estimate the valuations held by their counterparts. A common approach involves constructing a belief distribution.

**Definition 2** (Belief Distribution). *A belief system for agent $i$ is a probability mass function $\mathbb{B}_i$ defined over the set of all feasible valuations $\Omega$:*

$$\Omega := \big\{ \mathbf{v}_{-i} = (v_{-i}^1, v_{-i}^2, \ldots, v_{-i}^k) \in \mathbb{R}_{\geq 0}^k \mid \sum_{j=1}^{k} v_{-i}^j = \mathcal{V} \big\},$$

*where $0 \leq v_{-i}^j \leq \mathcal{V}$ for each item $j \in [k]$. We denote $\mathbf{V}_{-i}$ as the random vector with support over $\Omega$.*

Initially, $\mathbb{B}_i$ is assumed to be uniform, reflecting the agents' lack of information about the valuations of others. This distribution is subsequently updated based on evidence gathered at each round of the negotiation, presented next.

**Allocation Proposal Process.**  At each round, the agent $i$ searches for an allocation $L_i$ that maximizes their own utility, while maintaining that the allocation is envy free according to their belief distribution $\mathbb{B}_i$. This procedure can be formally defined as an optimization problem under chance constraint:

$$\max_{L_i} U_i(L_i), \tag{1}$$
$$\text{s.t. } P_{\text{EF}}(L_i; \mathbb{B}_i) > 0.$$

With discrete allocations, this objective can be maximized by enumerating all possible allocations that attain a non-zero envy-free probability, decomposed as follows:

$$\begin{aligned} P_{\text{EF}}(L; \mathbb{B}_i) &:= P(\text{Allocation } L \text{ is envy free}; \mathbb{B}_i) \\ &= P\left(U_i(L_i) \geq U_i(L_{-i}) \text{ and } U_{-i}(L_{-i}) \geq U_{-i}(L_i); \mathbb{B}_i\right) \\ &= \mathbb{E}_{\mathbf{v}_{-i} \sim \mathbb{B}_i}\left[\mathbb{1}(U_i(L_i) \geq U_i(L_{-i}) \text{ and } U_{-i}(L_{-i}) \geq U_{-i}(L_i))\right]. \end{aligned} \tag{2}$$

The expectation in Equation (2) can be approximated via Monte-Carlo samples drawn from the agent's belief distribution (Liu et al., 2024). We instead defer to our LLM agents to decide whether the allocation $L$ is envy free and maximizes self-interest.

**Allocation Proposal.** Upon identifying an optimal allocation according to Problem 1, the player proposes this allocation to the other player, which decides whether to accept this proposal or propose proposes a counter offer. Formally, this proposal procedure can be defined as follows:

$$\texttt{propose\_offer} : \mathcal{P}(K) \to \mathcal{O} \times \mathcal{P}(K),$$

where $\mathcal{P}(K)$ denotes the power set of resources $K$ that contains the set of all valid proposals, and $\mathcal{O}$ denotes the set of all possible outcomes consisting of three disjoint events $\{\mathcal{A}, \mathcal{R}_1, \mathcal{R}_2\}$. We let $\mathcal{A}$ denote the event of acceptance, wherein the negotiation concludes. On the other hand, a rational opponent satisfying Assumption 1 must reject the proposal for either of the following reasons:

- $\mathcal{R}_1$: The allocation $L$ is not envy free according to the opponent;

- $\mathcal{R}_2$: The allocation $L$ is envy free, but there exists an alternative allocation that provides the opponent with higher utility.

In the event of rejection, an agent must update their belief in order to refine their proposals.

**Bayesian Update.** If an allocation is rejected, it is essential to update our belief about the opponent's valuation vector $\mathbf{v}_{-i}$ to better inform future proposals. In what follows, we denote $\mathcal{R} = \mathcal{R}_1 \cup \mathcal{R}_2$ as the union of possible reasons for rejection. Then, the belief update formula (Cripps, 2018) for each possible $\mathbf{v}_{-i}$ is given by:

$$\begin{aligned}
\mathbb{B}_i(\mathbf{v}_{-i}) &= (1-\lambda)\mathbb{B}_i(\mathbf{v}_{-i}) + \lambda P(\mathbf{v}_{-i} \mid \mathcal{R}) \\
&= (1-\lambda)\mathbb{B}_i(\mathbf{v}_{-i}) + \lambda \frac{P(\mathcal{R} \mid \mathbf{v}_{-i})\mathbb{B}_i(\mathbf{v}_{-i})}{\sum\limits_{\mathbf{v}_{-j} \in \Omega} P(\mathcal{R} \mid \mathbf{v}_{-j})\mathbb{B}_i(\mathbf{v}_{-j})}
\end{aligned} \tag{3}$$

where $\lambda \in [0,1]$ is a hyperparameter of the update step, and the likelihood $P(\mathcal{R} \mid \mathbf{v}_{-i})$ represents the probability that the opponent rejects the allocation $L$ assuming the opponent's valuation is $\mathbf{v}_{-i}$. This likelihood depends on whether $L$ is acceptable and self-interest-maximizing for the opponent. We propose the following formula to the likelihood that satisfies Assumption 1.

$$P(\mathcal{R} = r \mid \mathbf{v}_{-i}) = \begin{cases} \frac{1}{1+\gamma}, & \text{Event } \mathcal{R}_1 : U_{-i}(L_{-i}) < U_{-i}(L_i)\}; \\ \frac{\gamma}{1+\gamma}, & \text{Event } \mathcal{R}_2 : U_{-i}(L_{-i}) \geq U_{-i}(L_i) \text{ and } \exists L' \text{ s.t. } U_{-i}(L'_{-i}) > U_{-i}(L_{-i}), \\ & \text{and } P_{\text{EF}}(L') > 0; \\ 0, & \text{Otherwise.} \end{cases}$$

Here, $\gamma \in [0,1]$ represents the probability that the opponent rejects the allocation in anticipation of a better offer, even when the current allocation is envy free. In actual implementation, as we do not know $\gamma$, thus we acknowledge that the opponent may reject an envy free allocation with any positive probability $\gamma = 1$. This approach simplifies the implementation while capturing the essential behavior that the opponent might anticipate a better deal.

In short, rejection occurs if $L$ is either an unacceptable allocation (not envy free) or acceptable but suboptimal allocations (envy free but not utility-maximizing). The opponent accepts $L$ only if it is both envy free and utility-maximizing given their valuations. An overview of our algorithm is presented in Algorithm 1.

**Remark 1.** *Such Bayesian update assumes the rationality of the opponent: the update relies on the assumption that the opponent's actions are consistent with rational behavior as defined by our model. Any deviation from rationality can lead to incorrect belief updates. Thus it can be unrobust to potential attacks such as deception.*

---

**Algorithm 1** Algorithm for the Allocation Game 1 with Two Players

---

1: **Input:** Private valuation vector $\mathbf{v}_i$ and a set of common resources $K$
2: **Output:** Final allocation $L_i$
3: **while** `True` **do**
4:     $L_i \leftarrow \arg\max_{L_i} U_i(L_i)$ s.t.$P_{\text{EF}}(L_i; \mathbb{B}_i) > 0$. `# Optimize problem` 1
5:     outcome, $L_{-i} \leftarrow$ `propose_offer`$(L_i)$
6:     **if** outcome $== \mathcal{A}$ **then**
      **return** $L_i$
7:     **end if**
8:     Update belief distribution $\mathbb{B}_i$ with Equation (3)
9: **end while**

---

### 5.3 Introduction to "Deal or No Deal"

"Deal or No Deal" is a representative game for incomplete-information resource allocation game. It is designed to facilitate research in developing AI agents capable of engaging in human-like negotiation dialogues. It consists of over 5,800 human-human negotiation dialogues collected via Amazon Mechanical Turk, with 1052 dialogues in the test dataset. Each negotiation involves three types of items – books, hats, and balls – with random quantities. Each participant is randomly assigned point values for each item type, summing up to 10, which are hidden from the other participant. Participants communicate through natural language to agree on how to divide the items to maximize their individual point totals, without revealing the true value systems during negotiation.

To comprehensively evaluate the negotiation performance of the LLM-based agents, we assess not only whether they reach a Nash Equilibrium, *i.e.*, whether an agreement is achieved, but also examine the fairness and effectiveness of the resulting distribution. For fairness, we adopt the concept of envy freeness.

### 5.4 Experiment Setting

To observe whether LLMs are capable of negotiation and whether our workflow design is effective, we evaluate multiple SOTA LLMs on the dataset. We choose the top-50 most difficult datapoints (44 out of the 50 such datapoints have an envy free allocation) that have an envy free allocation instead of the 526 cases to converse expense of experiment.

We define the difficulty of a datapoint by computing the $\ell_1$ distance between the real valuations of the two players:

**Definition 3** (Difficulty). *A datapoint d has a difficulty level defined by the $\ell_1$ norm of the difference between the players' valuation vectors:*

$$Difficulty(d) = -\|v_i - v_{-i}\| = -\sum_{k=1}^{3} |v_i^k - v_{-i}^k|$$

*The larger Difficulty(d) is, the more difficult the datapoint is.*

By selecting data points with the largest negative $\ell_1$ distances, we focus on negotiation scenarios where the players have very similar valuations for the items. Such cases are inherently more difficult because when both players value the items similarly, they are likely to desire the same items, leading to increased competition and potential conflict during negotiations. This similarity in valuations makes it more challenging to find allocations that are acceptable to both parties without significant concessions.

The validity of our difficulty definition is supported by empirical human performance data, as presented in Table 5. This table summarizes the outcomes of human negotiations across various difficulty levels, measured by the $\ell_1$ distance between the players' valuation vectors.

Table 5 illustrates several key trends:

| Difficulty($d$) | -2 | -3 | -4 | -5 | -6 | -7 | -8 | -9 | -10 | -11 |
|---|---|---|---|---|---|---|---|---|---|---|
| total number of datapoints | 13 | 27 | 57 | 85 | 108 | 133 | 177 | 189 | 210 | 217 |
| Agreement rate | 0.5385 | 0.5556 | 0.5614 | 0.6235 | 0.6574 | 0.6917 | 0.7119 | 0.7249 | 0.7381 | 0.7373 |
| envy free rate | 0.3077 | 0.4074 | 0.4035 | 0.4824 | 0.5463 | 0.6015 | 0.6441 | 0.6614 | 0.6810 | 0.6820 |
| Pareto optimal rate | 0.5384 | 0.4444 | 0.4385 | 0.4823 | 0.5277 | 0.5413 | 0.5310 | 0.5396 | 0.5523 | 0.5529 |
| envy free and Pareto optimal rate | 0.3077 | 0.3333 | 0.3333 | 0.3882 | 0.4537 | 0.4812 | 0.4858 | 0.4973 | 0.5142 | 0.5161 |

Table 5: Percentage of datapoints where humans achieve agreement, envy free allocations, pareto optimal allocations, and allocations that are both envy free and pareto optimal with different levels of difficulty.

**Agreement Rate:** The proportion of negotiations where participants reached an agreement increases with smaller difficulty levels. It starts at approximately 53.85% for a difficulty of -2 and rises to around 73.73% at a difficulty of -11. This suggests that when players value items differently (higher difficulty), they are more likely to reach an agreement, possibly because they have less overlap in their preferred items.

**Envy free Rate:** The percentage of negotiations resulting in envy free allocations also increase with smaller difficulty. It goes from about 30.77% at difficulty -2 to approximately 68.20% at difficulty -11. This indicates that as players' valuations diverge, it becomes easier to allocate items without causing envy.

**Pareto optimal Rate:** The rate of pareto optimal outcomes shows a slight increase as Difficulty($d$) becomes smaller. It starts at approximately 53.84% for Difficulty($d$) $= -2$ and fluctuates around the 55% mark at Difficulty($d$) $= -11$. This suggests that achieving pareto optimality becomes slightly more feasible as players' valuations diverge, although the trend is less pronounced compared to the agreement and envy free rates.

**Envy free and pareto optimal Rate:** The rate at which negotiations achieve both envy freeness and pareto optimality increases with less difficult datapoints. It starts from about 30.76% at Difficulty($d$) $= -2$ and increases to approximately 51.61% at Difficulty($d$) $= -11$. This trend reflects the combined effects observed in the individual envy free and pareto optimal rates.

These findings validate our difficulty definition by demonstrating that negotiations become more challenging for humans as the players' valuations become more similar (higher Difficulty($d$)). The lower agreement and envy free rates at higher difficulty levels underscore the increased difficulty in reaching mutually satisfactory agreements when players highly value the same items. Conversely, higher difficulty levels, indicating greater differences in valuations, facilitate agreements and fair allocations, as players are more willing to concede items they value less.

**LLM Backbone Models** To evaluate the negotiation performance of LLMs, we utilize four state-of-the-art LLMs as the backbone for agents in negotiation games: Claude-3.5 Sonnet (Sonnet), Claude-3 Opus (Opus), GPT-4o, and o1. For Claude-3.5 Sonnet, Claude-3 Opus, and GPT-4o, we set the temperature to 1.0 to encourage exploratory behavior in negotiation scenarios. For o1, we use the default temperature.

**Metrics** To comprehensively evaluate the performance of the LLM-based agents in the negotiation tasks, we employ several key metrics that capture different aspects of the negotiation process and outcomes. These metrics are designed to assess efficiency, individual utility, fairness, and overall effectiveness of the negotiations. The metrics are as follows:

- Number of Rounds: This metric represents the total number of dialogue exchanges (or turns) between the agents before reaching an agreement or terminating the negotiation.

- Agreement Percentage (Agreement): Whether agreement is achieved in the negotiation.

- Agent Score: The agent score measures the utility that an agent obtains from the final agreement. It is calculated based on the agent's own valuation of the items they receive.

- Pareto Optimality Percentage (PO): This metric determines whether the final allocation is pareto optimal

- Envy Freeness Percentage (EF): This metric assesses whether the final allocation is envy free

- Total Score: The total score is the sum of the utilities obtained by both agents from the final allocation

By analyzing these metrics, we aim to capture a comprehensive picture of the negotiation outcomes: effectiveness of negotiation is measured through the number of rounds and pareto optimality; individual utility maximization is measured via the agent scores; fairness is measured by checking for envy freeness; overall effectiveness is measured through pareto optimality and the total score.

## 5.5 Experiment Result

In this section, we present the evaluation results across three distinct settings: (1) agents operating without the workflow, which assesses the baseline negotiation capabilities of the LLMs; (2) agents utilizing the workflow, which evaluates the effectiveness of the proposed workflow in enhancing negotiation performance; and (3) a comparative analysis of individual agents functioning both with and without the workflow, highlighting the impact of the workflow on their performance. The results provide insights into the strengths and limitations of the workflow, as well as the inherent negotiation abilities of different LLMs.

### 5.5.1 Both Agents without Workflow

We present the results of our experiments involving four LLM-based agents—Claude-3.5 Sonnet, Claude-3 Opus, GPT-4o, and Model o1—alongside human performance provided by the original dataset and the best possible outcomes for the selected data points. By "best possible outcome," we refer to an allocation that is both pareto optimal and envy free while maximizing the total reward for both players. For each metric, we report the average scores across the 44 datapoints selected based on the difficulty metric defined earlier.

| Model | Negotiation Round | Agreement | Alice score | Bob score | PO | EF | total reward |
|---|---|---|---|---|---|---|---|
| Best | – | 1.0000 | 5.82 | 6.66 | 1.0000 | 1.0000 | 12.48 |
| Human | 2.86 | 0.6817 | 3.32 | 3.39 | 0.4317 | 0.4545 | 6.64 |
| Sonnet | 7.07 | 0.9545 | 5.55 | 5.57 | 0.7045 | 0.7045 | 11.11 |
| o1 | 3.86 | 0.7500 | 4.39 | 4.43 | 0.4545 | 0.4772 | 8.82 |
| GPT-4o | 18.45 | 0.6363 | 2.80 | 4.38 | 0.4091 | 0.3864 | 7.14 |
| Opus | 4.37 | 0.4772 | 2.68 | 3.02 | 0.3636 | 0.2727 | 5.70 |

Table 6: Raw-LLM vs. Raw-LLM

As shown in Table 6, for these 44 datapoints, it is always possible to find pareto optimal and envy free allocations that maximize the total reward for both players. However, **human performance falls significantly short of this ideal**. The human participants achieved an agreement rate of 68.17%, with an average total reward of 6.64. The percentages of negotiations resulting in pareto optimal and envy free allocations are 43.17% and 45.45%, respectively.

Among the LLM-based agents, **Claude-3.5 Sonnet demonstrates the best super-human performance**. It achieves an agreement rate of 95.45%, and its average total reward is 11.11, which is close to the best possible total reward of 12.48. The average scores forAlice and Bob are 5.55 and 5.57, respectively. However, the rates of achieving pareto optimality and envy freeness are 70.45% each, indicating that while the agent performs well in terms of reaching agreements and maximizing total rewards, there is still a gap in consistently achieving the most efficient and fair outcomes. Also notice that all LLMs can performance better than human baseline.

**Effect of Temperature** We also observed that the performance of LLM is highly sensitive to the temperature parameter used during generation (Krishnamurthy et al., 2024). To investigate this, we conducted full experiments with GPT-4o at temperatures of 0.0 and 1.0. The results are presented in Table 7.

| Model | Negotiation Round | Agreement | Alice score | Bob score | PO | EF | total reward |
|---|---|---|---|---|---|---|---|
| temp=0.0 | 19.36 | 0.5681 | 2.98 | 3.47 | 0.4091 | 0.3260 | 6.44 |
| temp=1.0 | 18.45 | 0.6364 | 2.80 | 4.38 | 0.4090 | 0.3864 | 7.14 |

Table 7: GPT-4o with temperature 0.0 and 1.0

These findings indicate that **the negotiation performance of LLMs is highly sensitive to the temperature parameter**, which influences the randomness and diversity of generated responses. Setting the temperature to 1.0 yields improvements in agreement rate, total reward, and envy freeness. This outcome may result from the increased exploration encouraged by a higher temperature, allowing the LLMs to consider a broader range of potential allocations that facilitate agreement. Based on this observed benefit, we selected a temperature setting of 1.0 for raw-LLM vs. raw-LLM experiments[2].

> **Summary: raw-LLM vs. raw-LLM Performance**
>
> - **Outstanding Performance of Claude-3.5 Sonnet**: Among all the evaluated LLMs, Claude-3.5 Sonnet exhibits the highest performance, achieving results that are close to the best possible outcomes in the negotiation games.
> - **Superhuman Capabilities of Sonnet and o1 Models**: Both Claude-3.5 Sonnet and model o1 demonstrate performance that surpasses human performance.
> - **Effect of Temperature on Exploration and Outcomes**: Employing higher temperature settings encourages greater exploration of possible strategies by the LLMs, leading to improved results.

### 5.5.2 Both Agents with Workflow

In this set of experiments, we employ the proposed negotiation workflow for both agents. We did not include Model o1 in our experiments for two main reasons: (1) the computational cost associated with running Model o1 is prohibitively high, and (2) preliminary experiments indicated that Model o1 does not perform optimally when utilizing external workflows.

| Model | Negotiation Round | Agreement | Alice score | Bob score | PO | EF | total reward |
|---|---|---|---|---|---|---|---|
| Best | - | 1.0000 | 5.82 | 6.66 | 1.0000 | 1.0000 | 12.48 |
| Opus | 4.05 | 1.0000 | 5.82 | 6.50 | **0.9091** | 0.9318 | **12.31** |
| GPT-4o | 4.91 | 1.0000 | 5.93 | 6.25 | 0.8636 | **1.0000** | 12.18 |
| Sonnet | 4.45 | 1.0000 | 5.93 | 6.16 | 0.7953 | 0.9772 | 12.11 |

Table 8: Workflow-LLM vs. Workflow-LLM

Several key observations emerge from the results:

**Reduced negotiation rounds:** The number of negotiation rounds required to reach an agreement is significantly reduced compared to previous experiments. This reduction indicates a much more effective and efficient negotiation process facilitated by the workflow.

**Universal agreement achievement:** Agreements were reached in all data points. This consistent success suggests that the agents are effectively reaching the Nash Equilibrium when utilizing the workflow.

**Increased pareto optimality rate:** The pareto optimality rates have increased substantially, with Claude-3 Opus obtaining the highest performance on achieving a pareto optimal deal. This improvement indicates that the workflow aids the agents in finding more efficient allocations where no player can be made better off without making the other worse off.

**High envy freeness rate:** The agents achieve near-perfect envy freeness rates. GPT-4o attains a 100%

---

[2]We did not experiment with o1 model due to extremely high cost

envy freeness rate, while Claude-3.5 Sonnet reaches 97.72%. This outcome demonstrates that the negotiated agreements are perceived as fair by both agents, aligning with the envy freeness criterion.

**Total rewards approaching optimal:** The total rewards obtained by the agents are now very close to the best possible total reward. Claude-3 Opus achieves the highest average total reward of 12.31, missing only 0.17 compared with the best possible result. This proximity to the optimal total reward highlights the effectiveness of the workflow in maximizing joint utility.

*Notice that interestingly, the model that performs the worst without workflow now performs the best with workflow, and the model that performs the best without workflow now performs the worst with workflow.*

**Effect of Temperature**  Furthermore, we observe that **LLMs employing the proposed workflow exhibit increased stability with respect to temperature**. To demonstrate this, we conducted a comprehensive experiment using GPT-4o with temperature values set to 0.0 and 1.0. The results are presented in Table 9.

| Model | Negotiation Round | Agreement | Alice score | Bob score | PO | EF | total reward |
|---|---|---|---|---|---|---|---|
| temp=0.0 | 4.80 | 1.0000 | 5.53 | 6.67 | 0.8695 | 1.0000 | 12.20 |
| temp=1.0 | 4.91 | 1.0000 | 5.93 | 6.16 | 0.8636 | 1.0000 | 12.18 |

Table 9: Workflow-GPT-4o with temperature 0.0 and 1.0

The performance metrics remain highly consistent between the two temperature settings. Both temperature values result in a 100% agreement rate and a 100% envy freeness rate. The total rewards are comparable, with 12.20 at temperature 0.0 and 12.18 at temperature 1.0. The number of negotiation rounds and the pareto optimality rates are also similar, indicating that the workflow enables the LLM to maintain stable performance regardless of temperature changes.

**Performance of Valuation Estimation**  We observe that the strong performance of the negotiation workflow is primarily due to the agents' accuracy in predicting the valuations of the other player. To quantify the accuracy of valuation estimation, we introduce three metrics:

**Precision**: This metric assesses whether the set of possible valuations, assigned a probability greater than zero after belief updating, includes the opponent's true valuation. Formally, let $\mathbf{V}_{est} = \{\mathbf{v}_{-i} \mid P(\mathbf{v}_{-i}) > 0\}$ be the set of estimated valuations with non-zero probability, and $\mathbf{v}_{-i}^{true}$ be the opponent's true valuation vector. Precision is defined as:

$$\text{Precision} = \mathbb{1}[\mathbf{v}_{-i}^{true} \in \mathbf{V}_{est}]$$

**Recall**: This metric measures the specificity of the estimated valuation set, indicating how many incorrect valuations are included alongside the true valuation. Recall is calculated as:

$$\text{Recall} = \frac{\mathbb{1}[\mathbf{v}_{-i}^{true} \in \mathbf{V}_{est}]}{|\mathbf{V}_{est}|}$$

where $|\mathbf{V}_{est}|$ is the cardinality of the estimated valuation set. A higher precision (*i.e.*, a smaller $|\mathbf{V}_{est}|$ signifies that the agent has narrowed down the opponent's valuation to a smaller set of possibilities, increasing the likelihood of accurate predictions.

**Reduction Percentage**: This metric evaluates how much the estimated valuation set has been reduced from the initial prior distribution. It is defined as:

$$\text{Reduction Percentage} = 1 - \frac{|\mathbf{V}_{est}|}{|\mathbf{V}_{prior}|}$$

where $|\mathbf{V}_{prior}|$ is the size of the initial prior valuation set before any belief updates. A higher reduction percentage indicates a significant narrowing of possible valuations, reflecting effective belief updating.

The average performance of the both agents in estimating the opponent's valuation is summarized in Table 10 across 44 datapoints. Notice that for all the three models, their estimations of the opponent's valuation are exactly the same, indicating the robustness of the workflow—no matter how the negotiation process goes, it can always compute the correct valuation.

| Model | Precision | Recall | Reduction Percentage |
|-------|-----------|--------|----------------------|
| Sonnet | 0.9545 | 0.3766 | 0.7033 |
| GPT-4o | 0.9545 | 0.3515 | 0.6980 |
| Opus | 0.7954 | 0.2737 | 0.6947 |

Table 10: Performance of Estimation of Valuation of the Other Player

The high recall values indicate that the agents are effective in ensuring the true opponent valuation remains within consideration throughout the negotiation. The high precision and reduction percentages demonstrate that the agents significantly narrow down the valuation space, although there remains room for improvement in eliminating incorrect valuations. To be more concrete on how accurate the valuation estimation is: a recall of 0.3766 means that, on average, the estimated set of possible valuations only contains approximately $\frac{1}{0.3766} = 2.66$ possible valuations.

To analyze how the belief distribution $\mathbb{B}_i(\mathbf{V_{-i}})$ changes throughout the negotiation process, we use Claude-3.5 Sonnet as an illustrative example. We examine the evolution of $\mathbb{B}_i(\mathbf{V_{-i}})$ over successive negotiation rounds by presenting the valuation estimation metrics—precision, recall, and reduction percentage—after each round. For each data point and negotiation round $n_r$, if $n_r$ exceeds the total number of negotiation rounds required for that data point, we utilize the results from the final round. In our dataset, the maximum number of negotiation rounds is 7.

| Metric | 1 | 2 | 3 | 4 | 5 | 6 | 7 |
|--------|---|---|---|---|---|---|---|
| Precision | 0.9545 | 0.9318 | 0.7500 | 0.8636 | 0.9318 | 0.9432 | 0.9545 |
| Recall | 0.2381 | 0.3099 | 0.2958 | 0.3079 | 0.3655 | 0.3652 | 0.3766 |
| Reduction Percentage | 0.5997 | 0.6825 | 0.7397 | 0.7011 | 0.7025 | 0.7033 | 0.7033 |

Table 11: Performance of Sonnet's Estimation of Opponent's Valuation Across Negotiations

Overall, we observe that with an increasing number of negotiation rounds, the recall and the reduction percentage tend to increase. This trend is intuitive because, as more rounds of negotiation occur, more information about the opponent's preferences is revealed. This additional information allows the agent to further narrow down the range of possible valuations for the opponent, resulting in a higher reduction percentage in the belief space $\mathbb{B}_i(\mathbf{V_{-i}})$.

*While recall and reduction percentage increase with more negotiation rounds, the precision exhibits a non-monotonic behavior — it initially decreases and then increases.* This pattern can be attributed to the nature of complex negotiations that require more rounds: In early rounds, the agent might eliminate certain valuations prematurely based on limited information, leading to a higher recall but potentially excluding the true valuation (lower precision). As negotiations progress, especially in cases where the agent initially made inaccurate estimations, additional information from the opponent's responses allows the agent to correct its beliefs. This correction process may temporarily increase the size of $\mathbf{V}_{est}^{n_r}$, decreasing precision. By the final rounds (up to a maximum of 7 in our data), the agent has accumulated sufficient information to refine its belief accurately, resulting in a convergence back to high precision and increased recall.

**Indistinguishable Set for Item Valuation**   It is noteworthy that the Opus model exhibits a relatively low precision in its estimated valuations – below 0.8 as presented in Table 10. Despite this, the Opus model achieves the highest performance in negotiation outcomes among all evaluated models. This apparent contradiction raises an intriguing question: *How can a model with imprecise valuation estimations attain superior negotiation results?*

This phenomenon can be explained by recognizing that, in resource allocation scenarios, different sets of item valuations can lead to the same optimal allocation. That is, multiple valuation profiles may belong to an *indistinguishable set*, wherein they result in identical optimal allocations, specifically the envy-free allocations that maximize total utility.

Consider a scenario involving two players and three types of items: one book, one hat, and three balls. Suppose the players' valuation vectors are as follows:

- First valuation: $\mathbf{v}_1 = (1, 3, 2), \mathbf{v}_{-1} = (1, 0, 3)$

- Second valuation: $\mathbf{v}_1 = (0, 4, 2), \mathbf{v}_{-1} = (1, 0, 3)$

In both scenarios, the optimal allocation, which we define as the envy free allocation that maximizes the total utility, is the same: $\text{Player}_1$ receives the hat and one ball, while $\text{Player}_{-1}$ receives the book and two balls. Formally, the allocations are $L_1 = \{\text{hat, ball}_1\}$, $L_{-1} = \{\text{book, ball}_2, \text{ball}_3\}$

Similarly, consider another pair of valuation vectors:

- Third valuation: $\mathbf{v}_1 = (1, 3, 2), \mathbf{v}_{-1} = (1, 0, 3)$

- Fourth valuation: $\mathbf{v}_1 = (0, 3, 2), \mathbf{v}_{-1} = (0, 1, 3)$

In both of these scenarios, the optimal allocation is that $\text{Player}_1$ receives 1 book, 1 hat, and 1 ball while $\text{Player}_{-1}$ receives 2 balls. Formally, the allocations are $L_1 = \{\text{book, hat, ball}_1\}$, $L_{-1} = \{\text{ball}_2, \text{ball}_3\}$.

These examples illustrate that different valuation profiles can lead to the same optimal allocation. Here, "optimal" refers to the envy free allocation that maximizes the total utility of all players.

An important implication of this observation is that an imprecise estimation of the opponent's valuation is not necessarily detrimental because different valuations may lead to the same best allocation. Specifically, for a given valuation $\mathbf{v}$, another valuation $\mathbf{v}'$ is considered *indistinguishable* from $\mathbf{v}$ if the set of best possible allocations (all envy free allocations with maximum total utility) based on $\mathbf{v}'$ is a subset of that based on $\mathbf{v}$. Formally, we define the *indistinguishable set* $\mathcal{I}(\mathbf{v})$ for a valuation profile $\mathbf{v}$ as follows:

**Definition 4** (Indistinguishable Set for One Item Valuation)**.** *Let $\mathbf{v}$ be a valuation profile for all players in the game, and let $\mathcal{L}^*(\mathbf{v})$ denote the set of optimal allocations under $\mathbf{v}$, specifically, all allocations that are envy free and maximize the total utility. The* indistinguishable set $\mathcal{I}(\mathbf{v})$ *for the valuation $\mathbf{v}$ is defined as:*

$$\mathcal{I}(\mathbf{v}) = \{\mathbf{v}' \mid \mathcal{A}^*(\mathbf{v}') \subseteq \mathcal{A}^*(\mathbf{v})\}$$

*That is, $\mathcal{I}(\mathbf{v})$ consists of all valuation profiles $\mathbf{v}'$ such that the set of optimal allocations under $\mathbf{v}'$ is a subset of the set of optimal allocations under $\mathbf{v}$.*

This definition formalizes the concept that different valuation profiles can be indistinguishable in terms of their implications for optimal allocations. From the perspective of a player, any valuation $\mathbf{v}'$ within the indistinguishable set $\mathcal{I}(\mathbf{v})$ does not necessitate a different strategic approach, as the optimal allocations remain consistent with those under their original valuation $\mathbf{v}$.

This concept is particularly useful in negotiation settings under incomplete information. It suggests that players may not need to precisely estimate the opponent's valuations if variations in those valuations lead to the same set of optimal allocations. By focusing on the indistinguishable set, players can simplify their strategic considerations and concentrate on reaching agreements that fall within the known optimal allocations, thereby facilitating more efficient and effective negotiations.

To evaluate the effectiveness of estimated valuations in capturing the indistinguishable set in the game, we compute the precision and recall of estimated valuations with respect to $\mathcal{I}(\mathbf{v}_{-i}^{true})$. Precision is defined as:

$$\text{Precision} = \mathbb{1}[\mathbf{V}_{est} \cap \mathcal{I}(\mathbf{v}_{-i}^{true}) \neq \emptyset]$$

indicating whether at least one estimated valuation falls within the indistinguishable set. Recall is defined as:

$$\text{Recall} = \frac{|\mathbf{V}_{est} \cap \mathcal{I}(\mathbf{v}_{-i}^{true})|}{|\mathbf{V}_{est}|}$$

representing the proportion of estimated valuations that are indistinguishable from the true valuation. We report the average precision and recall across all datapoints in Table 12:

| Model | precision w.r.t $\mathcal{I}(\mathbf{v})$ | recall w.r.t $\mathcal{I}(\mathbf{v})$ |
|---|---|---|
| Sonnet | 1.0 | 0.6022 |
| GPT-4o | 1.0 | 0.5633 |
| Opus | 1.0 | 0.5399 |

Table 12: Performance of estimated valuations with respect to indistinguishable of the true valuation.

The results indicate that, although precision varies among models in Table 10 and does not always achieve perfect precision, all models contain at least one valuation that is indistinguishable from the true valuation. Furthermore, the recall values are relatively high, suggesting that more than half of the valuations in the estimated sets are indistinguishable from the true valuation. This demonstrates that the models are effective in identifying valuations that lead to optimal allocations, even if they do not precisely estimate the opponent's exact valuations.

> **Summary for workflow-LLM v. workflow-LLM**
>
> - **Achievement of Near-Optimal Allocations**: When both agents utilize the workflow, all models consistently achieve allocations that are close to the best possible outcomes.
>
> - **Reversal in Performance Rankings**: The performance hierarchy observed without the workflow is reversed in this setting. Claude-3 Opus exhibits the highest performance, followed by GPT-4o and then Claude-3.5 Sonnet.
>
> - **Precision in Valuation Estimation**: The workflow-enhanced LLMs display remarkable precision in estimating the opponent's valuations. They effectively reduce the set of possible valuations to as few as 2 or 3 options and all of them are precise in recovering an indistiguishable estimated valuation for the true valuation. This strong performance indicates effective information flow in negotiations based on the workflow.

### 5.5.3 One Agent with Workflow

In this section, we present experimental results where only one LLM-based agent employs the proposed negotiation workflow, while the other agent negotiates using direct prompting without the workflow. Specifically, we conduct experiments in two scenarios:

- Workflow-LLM vs. Raw-LLM: OnlyAlice uses the workflow, and Bob uses direct prompting.

- Raw-LLM vs. Workflow-LLM: Only Bob uses the workflow, andAlice uses direct prompting.

An interesting phenomenon emerges from the results: the agent not using the workflow tends to achieve a higher individual reward than the agent using the workflow. This outcome can be attributed to the following factors: (1) The workflow-guided agent is designed to consider envy freeness from both its own perspective and the opponent's perspective. This consideration leads the agent to make more cooperative offers, potentially sacrificing some of its own utility to achieve fairness. (2) The non-workflow agent, lacking

| Model | Negotiation Round | Agreement | Alice score | Bob score | PO | EF | total reward |
|-------|-------------------|-----------|-------------|-----------|------|------|--------------|
| Sonnet | 6.91 | 0.9773 | 4.88 | 6.57 | 0.6136 | 0.5909 | 11.45 |
| GPT-4o | 11.84 | 0.8182 | 3.66 | 6.18 | 0.5909 | 0.3636 | 9.84 |
| Opus | 3.86 | 0.9091 | 5.09 | 5.53 | 0.6136 | 0.5909 | 10.52 |

Table 13: Workflow-LLM vs. Raw-LLM

| Model | Negotiation Round | Agreement | Alice score | Bob score | PO | EF | total reward |
|-------|-------------------|-----------|-------------|-----------|------|------|--------------|
| Sonnet | 6.45 | 1.0000 | 6.39 | 5.70 | 0.7727 | 0.5909 | 12.09 |
| GPT-4o | 11.36 | 0.8181 | 5.75 | 4.14 | 0.6136 | 0.5227 | 9.89 |
| Opus | 3.89 | 0.7955 | 4.86 | 4.57 | 0.4318 | 0.5455 | 9.43 |

Table 14: Raw-LLM vs. Workflow-LLM

such constraints, may act more self-interestedly, proposing allocations that favor itself without ensuring fairness.

As a result, when the workflow agent negotiates with a non-workflow agent, the workflow agent may have to accept less favorable allocations to reach an agreement, or negotiations may stall if the workflow agent rejects unfair offers from the non-workflow agent.

## 5.6 To adopt the workflow or not?

The experiment result in section 5.5.3 raises a critical game-theoretic question: is it rational for an agent to adopt the workflow when the opponent may not do the same? Should a player use the workflow given the potential for exploitation by a non-cooperative opponent? To address this, we represent the decision-making scenario using payoff matrices, where each agent has two strategic choices: (1) Use Workflow: Apply the proposed negotiation workflow, and (2) Do Not Use Workflow: Engage in direct prompting without the workflow. The payoff matrix for each model is presented in Table 15.

| Models | Sonnet | | GPT-4o | | Opus | |
|--------|--------|---------|--------|---------|------|---------|
| Actions | use | not use | use | not use | use | not use |
| use | 5.82, 6.16 | 4.88, 6.57 | 5.93, 6.25 | 3.66, 6.18 | 5.82, 6.50 | 5.09, 5.53 |
| not use | 6.39, 5.07 | 5.55, 5.57 | 5.75, 4.14 | 2.80, 4.38 | 4.86, 4.57 | 2.80, 4.38 |

Table 15: Payoff matrices for using the workflow or not for the three models

For these payoff matrices, each cell represents the payoffs ($u_{Alice}$, $u_{Bob}$) under the corresponding strategies. For instance, in the payoff matrix for GPT-4o-based agent: (1) When both agents use the workflow, the payoffs are (5.93, 6.25) (2) WhenAlice uses the workflow and Bob does not, the payoffs are (3.66, 6.18) (3) WhenAlice does not use the workflow and Bob does, the payoffs are (5.75, 4.14) (4) When both agents do not use the workflow, the payoffs are (2.80, 4.38).

**For Claude-3.5 Sonnet** Alice's dominant strategy is not to use the workflow. If Bob uses the workflow: thenAlice's payoff for using the workflow is 5.82 and that for not using the workflow is 6.39. Since 6.39 > 5.82,Alice's best response is to not use the workflow; If Bob does not use the workflow:Alice's payoff for using the workflow is 4.88 and that for not using the workflow is 5.55. Since 5.55 > 4.88,Alice's best response is again to not use the workflow. Thus "not using the workflow" is the dominant strategy forAlice. Assuming rational behavior, Bob will also choose to not use the workflow. IfAlice does not use the workflow, Bob's payoff for not using the workflow is 5.57, which is greater than his payoff of 5.07 when he uses the workflow (since 5.57 > 5.07). Consequently, Bob's best response is to not use the workflow whenAlice does not use it.

**The Nash Equilibrium in this setting occurs when both agents choose not to use the workflow, resulting in payoffs of 5.55 forAlice and 5.57 for Bob. This is a classic Prisoner's Dilemma situation, where the Nash Equilibrium is not pareto optimal.** If both agents were to use the workflow, they would receive higher individual payoffs – 5.82 forAlice and 6.16 for Bob – and a higher combined total reward of 12.48, compared to 11.12 when both do not use the workflow. This indicates that both agents could be better off by mutually adopting the workflow, achieving an outcome that is Pareto superior to the Nash Equilibrium.

The dilemma arises because, while mutual cooperation leads to a better collective outcome, each agent has an incentive to deviate unilaterally from the cooperative strategy to increase their individual payoff. To achieve the pareto optimal outcome where both agents use the workflow, additional mechanisms are necessary to align individual incentives with collective welfare. *One possible solution is to introduce a punishment strategy or enforceable agreements that discourage unilateral defection.* For example, implementing repeated interactions with memory of past behavior could promote cooperation through strategies like "tit-for-tat," where agents reciprocate the opponent's previous action. Such mechanisms can alter the payoff structure by imposing future costs on defection, thereby making cooperation the rational choice in the long run.

**For GPT-4o**  The situation differs from Claude-3.5 because here, the Nash Equilibrium strategy is to use the workflow, which can be deduced from Iterative elimination of dominated strategy.

Firstly,Alice's dominant strategy is to use the workflow. If Bob uses the workflow:Alice's payoff for using the workflow is 5.93 while for not using the workflow is 5.75. Since $5.93 > 5.75$,Alice should use the workflow. But if Bob does not use the workflow:Alice's payoff for using the workflow is 3.66 while for not using the workflow is 2.80. Since $3.66 > 2.80$,Alice should again choose to use the workflow. Thus, using the workflow is a dominant strategy forAlice, as it yields a higher payoff regardless of Bob's choice. While Bob's lack of workflow may allow him to obtain a slightly higher reward in some instances,Alice still achieves a greater overall benefit by employing the workflow. Knowing thatAlice's dominant strategy is to use the workflow, Bob can anticipateAlice's choice. AsAlice will adopt the workflow, then if Bob does not adopt the workflow, he would obtain payoff 6.18; if Bob adopts the workflow as well, he will obtain 6.25. As 6.25 is larger than 4.14, Bob should use the workflow. **Thus, the rational choice for both agents is to use the workflow. This choice represents a Nash equilibrium, where neither agent has an incentive to deviate unilaterally from the strategy of using the workflow. It is also the pareto optimal strategy.**

**For Claude-3 Opus**  **The situation is analogous to that of GPT-4o, wherein the Nash Equilibrium is to use the workflow.** Utilizing the workflow constitutes a dominant strategy for bothAlice and Bob.

Specifically,Alice's dominant strategy is to use the workflow, as her payoffs are superior regardless of Bob's choice: 5.82 versus 4.86 when Bob uses the workflow, and 5.09 versus 2.68 when Bob does not use the workflow. Similarly, Bob's dominant strategy is to use the workflow, as his payoffs are higher irrespective ofAlice's decision: 6.50 versus 5.53 whenAlice uses the workflow, and 4.57 versus 3.02 whenAlice does not use the workflow. Consequently, the Nash Equilibrium occurs when both agents choose to use the workflow, resulting in payoffs of 5.82 forAlice and 6.50 for Bob. This outcome is pareto optimal, as neither agent can improve their own payoff without diminishing the other's payoff. Unlike the Claude-3.5 Sonnet scenario, where mutual defection led to a suboptimal equilibrium, the Claude-3 Opus scenario demonstrates that aligned incentives and the adoption of the workflow can lead to mutually beneficial and efficient outcomes.

### 5.6.1   Comparison and Implications

The analyses of Claude-3.5 Sonnet, GPT-4o, and Claude-3 Opus reveal distinct strategic dynamics based on their respective payoff matrices:

- Claude-3.5 Sonnet: Exhibits a classic Prisoner's Dilemma structure where the dominant strategy for both agents is to defect (not use the workflow), leading to a Nash Equilibrium that is not pareto optimal.

- GPT-4o and Claude-3 Opus: Both present scenarios where using the workflow is a dominant strategy for both agents. This leads to a Nash Equilibrium that is also pareto optimal, where both agents achieve higher individual and combined payoffs compared to other strategy profiles. These cases demonstrate that when strategies are aligned and cooperation is incentivized, agents can achieve mutually beneficial outcomes.

Adopting the workflow consistently results in pareto optimal outcomes across different language models. However, the rationality of this choice – specifically, whether using the workflow constitutes a Nash Equilibrium—depends on the particular characteristics of each language model. In the case of Claude-3.5 Sonnet, which demonstrates superior performance when utilizing the workflow, the Nash Equilibrium is observed to be the decision not to adopt the workflow. This outcome may be attributed to Claude-3.5 Sonnet's advanced negotiation capabilities, whereby the implementation of the workflow inadvertently increases its susceptibility to exploitation. Although the workflow provides certain advantages, these benefits are insufficient to outweigh the risks associated with its adoption. Consequently, Claude-3.5 Sonnet opts to forgo the workflow to mitigate potential exploitation, despite the inherent benefits the workflow could offer. This suggests that while the workflow can enhance performance, its effectiveness is contingent upon the underlying negotiation strengths of the language model, highlighting the necessity for tailored strategies that align with each model's unique capabilities and vulnerabilities.

> **Summary for raw-LLM v. workflow-LLM and workflow-LLM v. raw-LLM**
>
> - **Exploitation of Workflow-Enhanced LLMs**: Empirical results indicate that LLMs operating without the workflow often achieve higher payoffs when interacting with workflow-enhanced LLMs. This suggests that the structured workflow can be exploited by opponents not utilizing it, potentially leading to suboptimal outcomes for the workflow-enhanced agents.
>
> - **Strategic Decision on Workflow Adoption**: The choice of whether to adopt the workflow itself constitutes a game-theoretic dilemma. The dominant strategy – opting to use or forgo the workflow – depends on the specific characteristics and strategic incentives of the LLM models involved. This underscores that the rational decision regarding workflow adoption is contingent upon the model's capabilities and the anticipated behavior of opponents.
>
> - **Meta-Strategy for Workflow Adoption**: The question of whether to adopt the workflow introduces a need for a meta-strategy. Agents must consider not only their immediate negotiation tactics but also the higher-level strategy of employing the workflow. This involves weighing the potential benefits of the workflow against the risks of exploitation by opponents who may not be using it. Developing an effective meta-strategy requires agents to assess the specific context, anticipate opponents' choices, and adapt their approach accordingly to maximize their own utility while mitigating vulnerabilities.

## 6 Conclusion

This study conducted a comprehensive game-theoretic analysis to evaluate the rationality and effectiveness of adopting a negotiation workflow within Large Language Models (LLMs) across a spectrum of classic strategic scenarios. By modeling interactions through well-established complete-information games, including the Prisoner's Dilemma, Stag Hunt, Battle of the Sexes, Wait-Go Game, Duopolistic Competition, Escalation Game, Monopoly Game, Draco and Harry Game, we assessed how different LLMs, specifically Claude-3.5 Sonnet, GPT-4o, and Claude-3 Opus, navigate the balance between cooperation and competition.

Expanding our investigation to more realistic settings, we explored the Deal-No-Deal Game, which incorporates incomplete-information, to assess whether LLMs can efficiently allocate resources and negotiate agreements under conditions of uncertainty. In this context, we designed a workflow based on Bayesian updates to achieve pareto optimal and envy free allocations, thereby enhancing the negotiation process.

Our findings indicate that the adoption of the workflow generally promotes pareto optimal outcomes, wherein both agents achieve higher collective payoffs compared to non-cooperative strategies. For instance, in scenarios like the Stag Hunt and Battle of the Sexes, the workflow facilitated effective negotiation and coordination, enabling LLMs to reach mutually beneficial equilibria. Similarly, in the Deal-No-Deal Game, the structured workflow enhanced decision-making processes, leading to more efficient resource allocations.

## 7  Future Directions

This study opens several promising avenues for future research:

**Exploration of Workflow Vulnerabilities and Defense Mechanisms:**  Investigating how negotiation workflows can be exploited through methods such as deception is crucial, particularly in contexts like the Deal-No-Deal Game. Understanding the potential vulnerabilities within these workflows will enable the development of robust defense strategies to mitigate exploitation risks, thereby enhancing the reliability and security of negotiation protocols.

**Strategizing in Multi-Stage Games:**  Extending the analysis to multi-stage games introduces additional complexity, as the Nash Equilibrium may differ significantly from that in single-stage games. Future work should focus on how LLMs can effectively strategize over multiple stages, accounting for the dynamic evolution of the game state and the opponent's actions. This includes developing algorithms that can anticipate future moves and adjust strategies accordingly to maintain rationality and optimize outcomes.

**Development of Meta-Strategy:**  LLMs should be equipped with the capability to determine when to employ a particular workflow and when to adapt or forgo it. This necessitates the creation of a meta-strategy or meta-workflow that guides the selection of appropriate negotiation strategies based on the specific context, the agent's own abilities, the stage of the game, and the behavior of the opponent. Implementing such a meta-strategy would enhance the adaptability and effectiveness of LLMs across diverse negotiation scenarios.

**Alignment with Agent Interests and Stance Adoption:**  Currently, LLMs are generally aligned to be helpful and honest, lacking a personalized stance or specific interests. To function as proficient negotiation agents, it is imperative for LLMs to learn and adopt stances that reflect the interests they represent. This involves training LLMs to understand and advocate for particular objectives, thereby balancing their general alignment with the capacity to pursue defined goals within negotiations. Developing methods to instill a clear understanding of their own interests will enhance the LLMs' ability to engage in strategic decision-making and achieve desired outcomes.

These future directions aim to refine the strategic reasoning and negotiation capabilities of LLMs, ensuring they can operate effectively and rationally in complex, real-world scenarios. By addressing these areas, we can advance the development of LLMs as robust agents capable of navigating intricate strategic environments while safeguarding against potential vulnerabilities.

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

# A  Appendix

## A.1  Preliminary for Game Theory

In this section, we introduce fundamental notations, concepts, and definitions essential for understanding game theory and the strategic behavior of rational agents (Von Neumann & Morgenstern, 2007; Luce & Raiffa, 2012).

Consider a game involving $n$ agents, indexed by $i \in \mathbf{N} = \{1, 2, \ldots, n\}$. Each agent begins by interpreting the game's introduction and rules, which include detailed descriptions of the payoff matrices. Each agent has an action set defined as:

$$A_i = \{a_i^1, \ldots, a_i^k\}$$

where $k$ is the number of available actions for each player. The agents are also given the payoff matrix

$$U_i(\mathbf{a})$$

for all strategy profiles $\mathbf{a} = (a_1, a_2, \ldots, a_n)$ for $a_i \in A_i$. If there are only two players, we use $i \in \{1, -1\}$ to denote the two players and the payoff matrix is

$$U_i(a_i, a_{-i})$$

for all $a_i \in A_i$ and $a_{-i} \in A_{-i}$, where $A_{-i}$ is the action set of the other player.

We now present key definitions that will be utilized throughout this study.

**Definition 5** (Complete-information Game)**.** *A* complete information game *is a strategic game where all players have full knowledge of the game's structure, including the set of players, the action sets, and the payoff functions of all players. Specifically, each player knows:*

- *the total number of players involved in the game*

- *$A_j$: The set of actions available to each player $j$*

- *$U_j(a_1, a_2, \ldots, a_n)$: The payoff function for each player $j$, which assigns a real number to every possible strategy profile $(a_1, a_2, \ldots, a_n)$*

*This comprehensive knowledge allows each player to make informed strategic decisions, anticipating the choices and payoffs of other players based on complete information.*

**Definition 6** (Incomplete-information Game)**.** *An* incomplete information game *is a strategic game where there exists at least one player $i \leq n$ who does not have complete information about the payoff functions or action sets of other players $j \leq n$.*

*In incomplete information games, players form beliefs about the unknown elements based on available information and update these beliefs according to Bayesian principles as the game progresses. The lack of complete information requires players to strategize under uncertainty, considering not only the possible actions of others but also their possible types and the likelihood of various game structures.*

**Definition 7** (Simultaneous Game)**.** *A* simultaneous game *is a type of game where all players make their decisions or choose their actions at the same time, without knowledge of the choices made by the other players. In such games, players act independently and cannot coordinate their strategies based on others' actions.*

**Definition 8** (Sequential Game)**.** *A* sequential game *is a game in which players make decisions or choose actions in a specific order, with later players having some knowledge about earlier players' actions. These games are often represented using game trees and are analyzed using techniques like backward induction to determine optimal strategies.*

**Definition 9** (Payoff Matrix)**.** *A* payoff matrix *is a tabular representation used in game theory to illustrate the payoffs or utilities that each player receives for every possible combination of strategies chosen by all players in a game.*

*In a two-player game, let $A_1 = \{a_1^1, a_1^2, \ldots, a_1^m\}$ be the set of strategies available to Player$_1$, and $A_{-1} = \{a_{-1}^1, a_{-1}^2, \ldots, a_{-1}^n\}$ be the set of strategies available to Player$_{-1}$. The payoff matrix $U$ is an $m \times n$ matrix where each entry $U_{ij}$ corresponds on the strategy profile $(a_1^i, a_{-1}^j)$ and contains the payoff vector $(u_1(a_1^i, a_{-1}^j), u_{-1}(a_1^i, a_{-1}^j))$.*

*Informally, the payoff matrix is typically structured as follows:*

- *Rows represent the possible actions or strategies available to Player$_1$ (the* row *player).*

- *Columns represent the possible actions or strategies available to Player$_{-1}$ (the* column *player).*

- *Cells within the matrix contain ordered pairs of numbers $(u_1, u_{-1})$ where $u_1$ is the payoff to Player$_1$ and is the payoff to Player$_{-1}$ for the corresponding combination of strategies.*

**Definition 10** (Nash Equilibrium). *A Nash Equilibrium is a strategy profile in a game where no player can unilaterally improve their payoff by deviating from their current strategy, assuming the other players' strategies remain unchanged. Formally, a strategy profile $(a_1^*, a_2^*, \ldots, a_n^*,)$ is a Nash Equilibrium if, for every player i:*

$$U_i(a_i^*, a_{-1}^*) \geq U_i(a_i, a_{-1}^*)$$

*or all $a_i \in A_i$, where $a_{-1}^*$ represents the equilibrium strategies of all players other than player i.*

**Definition 11** (Envy freeness). *Envy freeness is a criterion for fair division that states that when resources are allocated to people with equal rights, each person should receive a share that they believe is at least as good as the share received by any other person.*

*An allocation is said to be* envy free *if no agent prefers another agent's allocation over their own. Formally, in an allocation among n agents, allocation $L = (L_1, L_2, \ldots, L_n)$ is envy free if for every pair of agents i and j,*

$$U_i(L_i) \geq U_i(L_j)$$

*where $U_i(L_k)$ denotes the utility of agent i for the allocation $L_k$ received by agent k.*

**Definition 12** (pareto optimality). *A strategy profile $\mathbf{a} = (a_1, a_2, \ldots, a_n)$ is considered* Pareto optimal *(or Pareto efficient) if there is no other feasible action profile $\mathbf{a}' = (a_1', a_2', \ldots, a_n')$ such that*

$$U_i(\mathbf{a}') \geq U_i(\mathbf{a})$$

*for all agents $i \in \{1, 2, \ldots, n\}$, and $U_j(\mathbf{a}') > U_j(\mathbf{a})$ for at least one agent j.*

*Therefore, an action profile $\mathbf{a}$ is Pareto optimal if it is impossible to make any agent better off without making at least one other agent worse off. This concept focuses on the efficiency of the collective action choices, ensuring that no improvement in one agent's payoff can be achieved without a detriment to another agent's payoff, as represented in the payoff matrix.*

## A.2 Details of Complete-information Games

**Prisoner's Dilemma** is a canonical example in game theory illustrating why two rational individuals might not cooperate, even when cooperation appears to be in their best interest. The game is structured as follows: Two suspects are arrested for a joint crime and interrogated separately, preventing communication. Each suspect has two possible actions: Cooperate and Defect. The payoffs for different actions are:

- If both cooperate, they each receive a light sentence (e.g., 1 year).

- If one defects and the other cooperates, the defector goes free (0 years), and the cooperator receives a heavy sentence (e.g., 5 years).

- If both defect, they each receive a moderate sentence (e.g., 3 years).

The Nash Equilibrium of this famous game is (Defect, Defect) where both players choose to defect each other. Notice that the Nash Equilibrium is not a Pareto optimal strategy, and the Pareto optimal strategy here is to cooperate with each other. The payoff matrix adopted in the paper is presented in Table 16a.

**Stag Hunt** represents a game of coordination and mutual trust. Two hunters decide whether to collaborate to hunt a stag or act individually to hunt a hare. Each hunter has two possible actions: Hunting a Stag and Hunting a hare. However, to successfully hunt a Stag, it requires both hunters to cooperate and then they each receive a high payoff; to successfully hunt a Hare, it does not require cooperation and each hunter can do it individually, while yielding a lower payoff. The payoff matrix adopted in the paper is presented in Table 16b.

There are two Nash Equilibria: (Stag, Stag) and (Hare, Hare). The (Stag, Stag) Nash Equilibrium is Pareto optimal but requires mutual cooperation and trust.

|  | Cooperate | Defect |
|---|---|---|
| Cooperate | 3, 3 | 0, 5 |
| Defect | 5, 0 | 1, 1 |

Table 16a: Payoff matrix for Prisoner's Dilemma

|  | Stag | Hare |
|---|---|---|
| Stag | 3, 3 | 0, 1 |
| Hare | 1, 0 | 1, 1 |

Table 16b: Payoff matrix for Stag Hunt

**Battle of the Sexes** is a coordination game involving two players Alice and Bob with different preferences over two possible activities but a shared desire to be together. Alice prefers opera; Bob prefers football, and both prefer attending the same activity over different ones. There are thus 2 possible actions for each player: going to opera and going to football. The payoff matrix adopted in the paper is presented in Table 17a.

There are two Nash Equilibria here: (Opera, Opera) and (Football, Football). Coordination is required to achieve these equilibria, but there can be a conflict over which equilibrium to select.

**Wait-Go Game** involves two drivers at an intersection deciding whether to go or wait. Each driver has two options of action: waiting which incurs a small cost of waiting but avoids collision, and risking a collision if the other driver also goes. The Nash Equilibria are the asymmetric strategies where one driver goes, and the other waits. The payoff matrix adopted in the paper is presented in Table 17b.

|  | Opera | Football |
|---|---|---|
| Opera | 2, 1 | 0, 0 |
| Football | 0, 0 | 1, 2 |

Table 17a: Payoff matrix for Battle of Sexes

|  | Wait | Go |
|---|---|---|
| Wait | 0, 0 | 0, 2 |
| Go | 2, 0 | -4,-4 |

Table 17b: Payoff matrix for Wait-Go Game

**Duopolistic Competition: Simple Cournot Competition** is a fundamental concept in industrial organization and game theory, examining how firms compete in markets with a small number of producers. One classic model to study such competition is the Cournot competition model, where firms compete on the quantity of output they decide to produce, and each firm's output decision affects the market price.

In the Simple Cournot Duopoly, there are two firms, Firm A and Firm B, producing a homogeneous product. Each firm independently chooses the quantity of output to produce, which determines the market price based on the total quantity supplied to the market, which in turn determines the profit. The strategic interdependence arises because each firm's optimal output depends on its expectations about the other firm's output. The Cournot-Nash Equilibrium occurs when neither firm can unilaterally change its output to increase its profit, given the output of the other firm.

Here, we adopt a scenario where there are 6 different possible actions of the two firms. The payoff matrix adopted in the paper is presented in Table 18. The Nash Equilibrium is (action 3, action 3) where both players can obtain reward 6. Notice that this Nash Equilibrium is not Pareto optimal, as the Pareto optimal strategy in the game is (action 2, action 2) where each player can obtain reward 7.

**Escalation Game** is a sequential game that models situations where two countries face decisions about escalating or de-escalating a conflict. Escalation may lead to higher potential payoffs but also increases the

|          | action 1 | action 2 | action 3 | action 4 | action 5 | action 6 |
|----------|----------|----------|----------|----------|----------|----------|
| action 1 | 0, 0     | 0,  9    | 0, 14    | 0, 15    | 0, 12    | 0,  5    |
| action 2 | 9, 0     | 7,  7    | 5, 10    | 3,  9    | 1,  4    | -1, -5   |
| action 3 | 14, 0    | 10,  5   | 6,  6    | 2,  3    | -2, -4   | -2, -5   |
| action 4 | 15, 0    | 9,  3    | 3,  2    | -3, -3   | -3, -4   | -3, -5   |
| action 5 | 12, 0    | 4,  1    | -4, -2   | -4, -3   | -4, -4   | -4, -5   |
| action 6 | 5, 0     | -5, -1   | -5, -2   | -5, -3   | -5, -4   | -5, -5   |

Table 18: A payoff matrix for Duopolistic Competition

risk of significant losses if both parties choose to escalate. Through the game, there are 2 action choices for each player: escalate and de-escalate.

In this game, two countries Country A and Country B, make decisions in a specific sequence and the payoff of each choice is presented in the tree structure payoff representation below:

```
Alice_choice_1: [0,0],
Alice_choice_2:
{
    Bob_choice_1: [1,-2],
    Bob_choice_2: {
        Alice_choice_1: [-2,1],
        Alice_choice_2: [-1,-1]
    }
}
```

**Monopoly Game**   is a sequential-move game that models the strategic interaction between a potential market entrant Company E and an incumbent monopolist Company I. It captures the dynamics of entry deterrence and the incumbent's decision to accommodate or fight the entrant. For Company E, there are two choices: staying out and entering market; For Company I, there are two choices, to accommodating Company E and fighting against Company E.

```
Alice_choice_1: [0,2],
Alice_choice_2:
{
    Bob_choice_1: [2, 1],
    Bob_choice_2: [-1, -1]
}
```

**Hot-cold Game**   is a sequential-move game that models the strategic interaction. Both players Alice and Bob have two choices on each stage.

```
Alice_choice_1:
{
    Bob_choice_1: [3, 2],
    Bob_choice_2: [2, 3]
},
Alice_choice_2:
{
    Bob_choice_1: [1, 4],
    Bob_choice_2: [4, 1]
}
```

**Draco Game**   is a sequential-move game with two players Alice and Bob with three stages. For each stage, there are two choices to make.

```
Alice_choice_1:
    {
        Bob_choice_1: [5, 5],
        Bob_choice_2:
            {
                Alice_choice_1: [2, 2],
                Alice_choice_2: [3, 4]
            }
    },
Alice_choice_2:
    {
        Bob_choice_1: [4, 5],
        Bob_choice_2:
            {
                Alice_choice_1: [5, 3],
                Alice_choice_2: [2, 2]
            }
    }
```

**TriGame**   is a sequential-move game with three stages. On each stage, the two players Alice and Bob have two choices.

```
Alice_choice_1:
{
    Bob_choice_1:
        {
            Alice_choice_1: [20, 3],
            Alice_choice_2: [0, 4]
        },
    Bob_choice_2:
        {
            Alice_choice_1: [2, 5],
            Alice_choice_2: [3, 4]
        }
},
Alice_choice_2:
{
    Bob_choice_1:
        {
            Alice_choice_1: [1, 5],
            Alice_choice_2: [4, 10]
        },
    Bob_choice_2:
        {
            Alice_choice_1: [2, 1],
            Alice_choice_2: [3, 2]
        }
}
```

## A.3 Detailed Observation on LLM's Rationality

This section provides an additional, comprehensive analysis of the rationality exhibited by LLM-based agents from various perspectives. Using Claude-3 Opus as a representative example, we examine the rational decision-making capabilities of LLMs in single-stage games through the following aspects:

1. Consistency of action choices across variations in payoff matrices: By analyzing the agents' decisions in different payoff matrix scenarios, we can determine whether the LLM-based agents maintain consistent action choices or if their choices are influenced by nuanced changes in the payoff matrix.

2. Consistency of action choices across designated personalities in system prompts: We investigate the impact of assigned personalities in system prompts on the agents' decision-making process. This analysis helps us understand whether the LLM-based agents' rationality is affected by the designated personality or if they maintain consistent action choices regardless.

3. Maintenance of rationality under discussion and multi-round discussions: We explore how the agents' rationality evolves when engaged in discussions or multi-round interactions. This examination reveals whether the LLM-based agents can maintain their rationality or if it is influenced by the communication and negotiation processes.

## A.4 Experiment Setup

In each experimental setup, we conduct 10 iterations utilizing a temperature value of 1 for the Claude-3 model. Each player within these setups is represented by an LLM-based agent. For each configuration, we document the probability distribution of the actions executed by these agents.

## A.5 Variance of Payoff Matrix

**Definition 13** (Nash-Equilibrium invariant perturbation). *A Nash Equilibrium Invariant Perturbation is a modification to the numerical values within a game's payoff matrix that preserves the set of Nash equilibria. Formally, consider a finite game $G = (N, \{S_o\}_{i \in N}, \{\pi_i\}_{i \in N})$ where $N$ is the set of players, $S_i$ is the strategy set for player $i$, $\pi_i : S \to \mathcal{R}$ is the payoff function for player $i$.*

*A perturbed game $G' = (N, \{S_i\}_{i \in N}, \{\pi'_i\}_{i \in N})$ of the game $G$ is defined by adjusted payoff functions $\pi'_i = \pi_i + \delta_i$ where $\delta_i : S \to \mathcal{R}$ represents the change in payoffs for player $i$. The perturbation $\delta = \{\delta_i\}_{i \in N}$ is termed a Nash Equilibrium Invariant Perturbation if the set of Nash equilibria remains unchanged between the original game $G$ and the perturbed game $G'$.*

Traditional evaluation of LLM on rationality utilize traditional game-theoretic scenarios. If LLMs do have rationality, then their rationality should be consistent across different instantiation of the payoff matrix.

To study this, we use the traditional game of Prisoner's Dilemma and Stag Hunt.

We introduce certain modifications to the traditional payoff matrices while ensuring that the rational choices remain unaltered. Despite these variations, the Nash Equilibrium remains consistent across all modifications. Consequently, if an agent is rational, it is expected to make the same rational choices in each scenario. We present the variations here:

|          | action 1   | action 2 |
|----------|------------|----------|
| action 1 | 300, 300   | 0, 301   |
| action 2 | 301, 0     | 1, 1     |

Table 19a: Variation 1: payoff matrix for Prisoner's Dilemma

|          | action 1 | action 2    |
|----------|----------|-------------|
| action 1 | 3, 3     | -300, 5     |
| action 2 | 5, -300  | -299, -299  |

Table 19b: Variation 2: payoff matrix for Prisoner's Dilemma

We conduct experiments on these two games with their variations. Figures 5 and 6 contain experiment results of the action distribution. Contrary to the expectation that performance should remain unaffected by payoff variations, the results demonstrate inconsistent performance distribution across different payoff

|          | action 1   | action 2 |
|----------|------------|----------|
| action 1 | 300, 300   | 0, 1     |
| action 2 | 1, 0       | 1, 1     |

Table 20a: Variation 1: payoff matrix for Stag Hunt

|          | action 1   | action 2   |
|----------|------------|------------|
| action 1 | 3, 3       | -100, -99  |
| action 2 | -99, -100  | -99, -99   |

Table 20b: Variation 2: payoff matrix for Stag Hunt

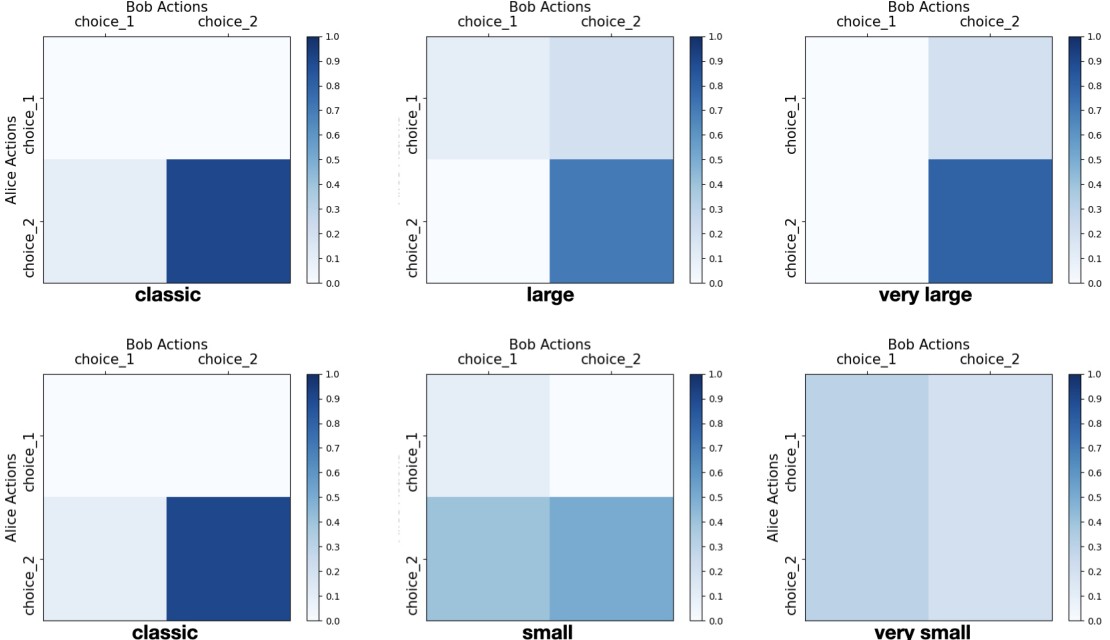

Figure 5: Agents' performance under different payoff matrix for Prisoner's Dilemma

scenarios. In the case of Prisoner's Dilemma, the probability of the rational situation (Action 2, Action 2) is significantly higher in the classic payoff matrix but considerably lower in the two variations. Similarly, in Stag Hunt, the actions taken also vary across different payoff scenarios. These findings suggest that the LLMs are either (1) not consistently making rational decisions or (2) their rationality is heavily influenced by other irrational factors.

## A.6 Variation of Personality

In addition to investigating the impact of payoff variations on the rationality of agents, we also examine whether the personality denoted in the system prompt influences the agents' rationality. If the agents are consistent in their computation and calculation of the reward, then their rationality should not be affected by the assigned "personality". The system prompt template used for this experiment is as follows:

```
You are a {{personality}} assistant that carefully answer the question.
```

For the personality variable, we select six different but common adjectives: compassionate, friendly, helpful, pragmatic, rational, and witty. The results of this experiment are presented in Figure 7. The findings indicate that the agents' performance varies significantly according to the assigned personality. The "witty" personality yields the game-theoretically most rational option, while the "rational" personality performs slightly worse. For other personalities, such as "compassionate", "friendly", "helpful", and "pragmatic", the agents exhibit decreased rationality and frequently make irrational choices.

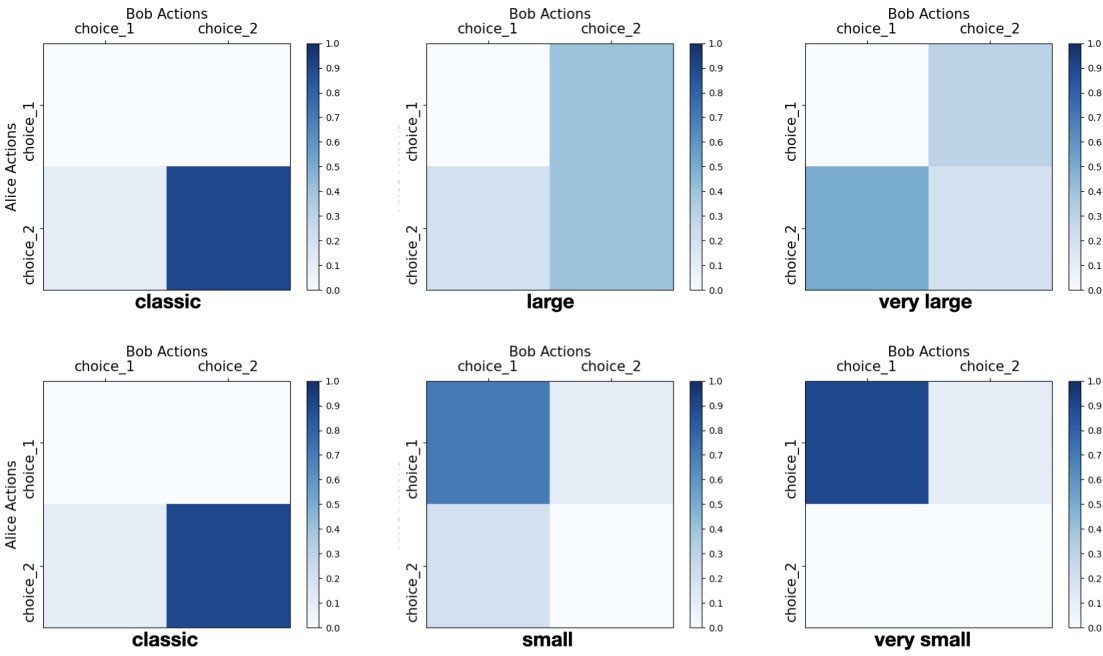

Figure 6: Agents' performance under different payoff matrix for Stag Hunt

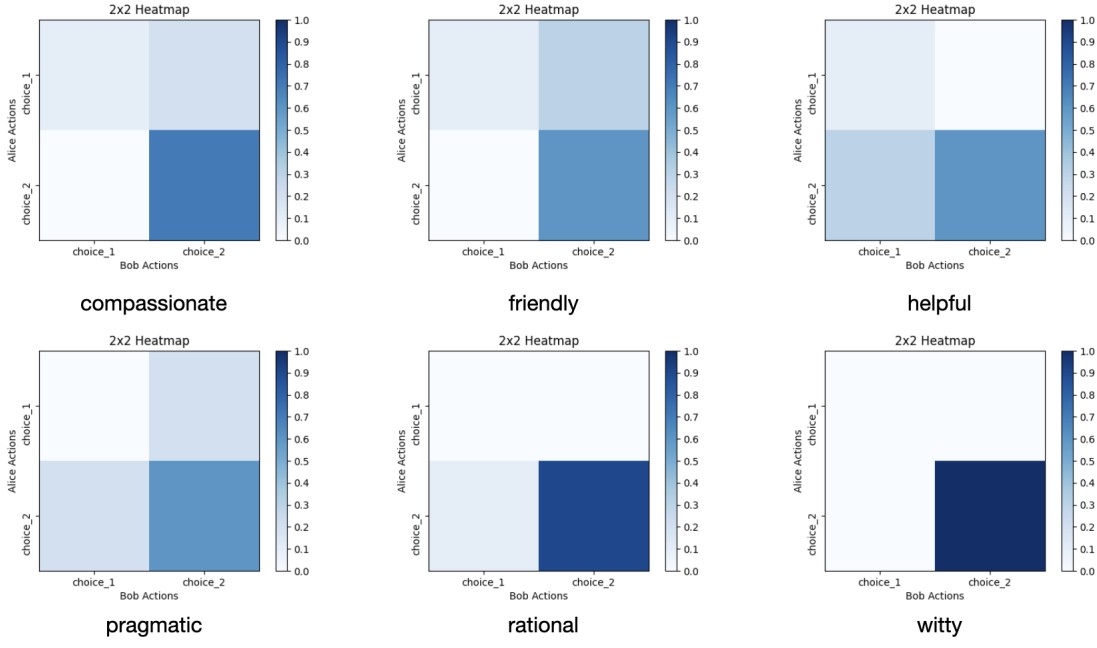

Figure 7: Agents' performance under different system prompt with personality

## A.7    Does negotiation affect rationality?

In certain situations, the decision-making process of agents can be influenced by discussions with other agents. For instance, in the game of Stag Hunt, effective negotiation can foster trust between players, enabling them

to recognize that individual efforts to capture a hare are not as advantageous as cooperating to hunt a stag. Consequently, successful communication should encourage players to select the pareto optimal Nash equilibrium instead of either of the two available equilibriums. However, there are scenarios where negotiation does not impact the outcome. In the case of Prisoner's Dilemma, communication fails to establish trust, as each player's dominant strategy is to defect regardless of the other player's action. Therefore, communication is unlikely to affect performance, as the rational choice remains unchanged.

In our current experimentation, we focus on three game-theoretic scenarios: Stag Hunt, Battle of the Sexes, and Prisoner's Dilemma. The impact of negotiation on these games varies as follows:

- Stag Hunt: In this game, negotiation plays a significant role in achieving the pareto optimal rational choice. Effective communication between players can foster trust and encourage cooperation, leading to better outcomes for both parties.

- Battle of the Sexes: Negotiation is crucial for enhancing coordination between the two players in this game. By discussing their preferences and intentions, players can reach a more coherent and mutually beneficial strategy.

- Prisoner's Dilemma: The presence of a single Nash Equilibrium in this game renders negotiation irrelevant. Regardless of any discussion, the rational choice for both players remains the same, and the outcome is determined by their individual decisions.

To investigate the impact of communication on agents' choices, we conduct experiments on both games with and without communication. In the communication setup, agents are allowed to exchange messages before making decisions. We record the action distribution of agents in each setup and present the results in Figure 8 for 0, 1, and 2 rounds of negotiation.

Based on our observations, we have identified several inconsistencies between our expectations and the actual outcomes in the various game-theoretic scenarios:

- Stag Hunt (Figure 8 (b)): As expected, negotiation leads to the adoption of the pareto optimal strategy. This result demonstrates that effective communication can facilitate cooperation and achieve better outcomes for all parties involved.

- Battle of the Sexes: (Figure 8 (c)): Our findings indicate an increase in coordination between players as the number of negotiation rounds increases. This outcome suggests that negotiation plays a crucial role in enhancing mutual understanding and promoting more coherent strategies.

- Prisoner's Dilemma (Figure 8 (a)): Contrary to expectations, players consistently gravitate towards the dominated strategy after negotiation. This result is surprising, as the presence of a single Nash Equilibrium should render negotiation irrelevant.

In summary, our findings suggest that negotiation does not always lead to the pareto optimal decision-making, and in some cases, it may even result in the loss of rationality. Further investigation is necessary to understand the underlying factors contributing to these unexpected outcomes.

### A.8 How does prompt affect the influence of negotiation?

Some people may want to ask can we use simple prompt engineering to control the effect of negotiation.

To investigate whether the observed negative impact of negotiation on player rationality can be mitigated through simple prompt engineering, we design six distinct prompts. The first three prompts emphasize caution regarding the other player's statements and encourage the agent to critically evaluate trustworthiness. These prompts are intended to foster a more skeptical and analytical mindset during negotiations. The remaining three prompts are formulated as commands, instructing the agent to make decisions independently without being influenced by the negotiation process. The objective of these prompts is to promote autonomy

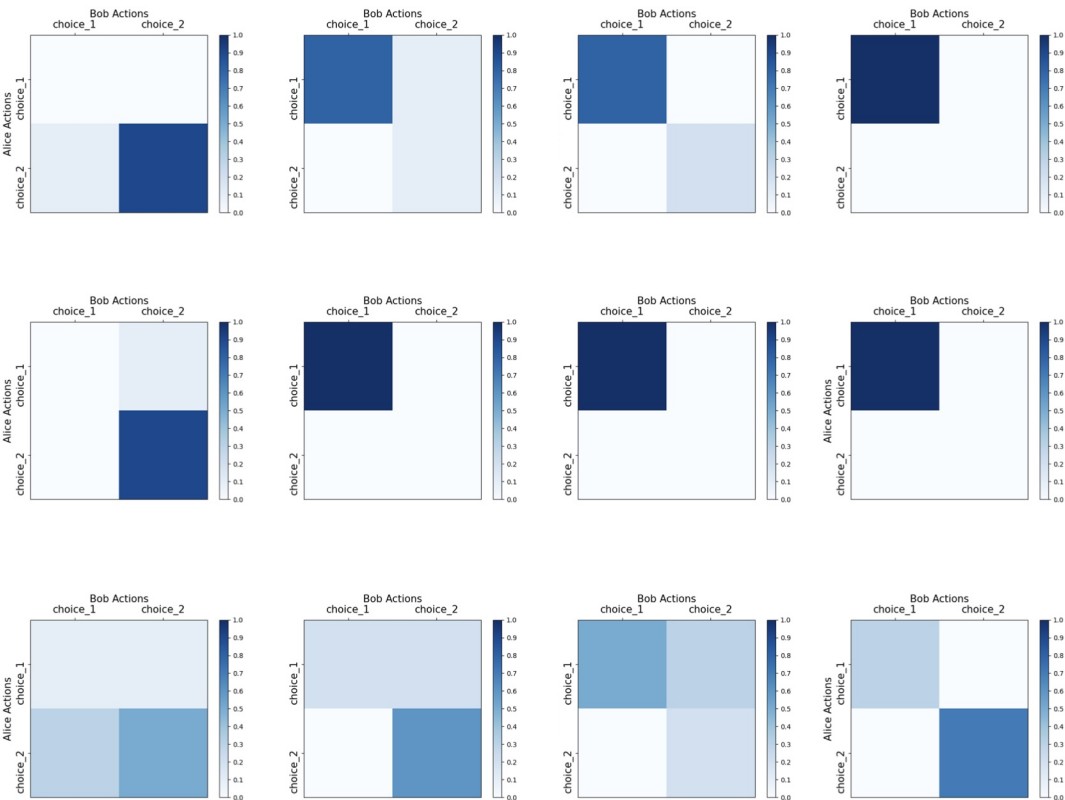

Figure 8: Agents' performance under different numbers of negotiation: 0-round, 1-round, 2-round, and 3-round from left to right for the four games

and self-reliance in the agent's decision-making, potentially minimizing the negative effects of negotiation on rationality.

In our experiment, we incorporate the six designed prompts at the end of the entire action-making prompt for each game. This placement ensures that the prompts serve as a final reminder or instruction to the LLM-based agents, not affected by the length of the negotiation texts.

```
Prompt 1: Please carefully analyze the negotiation messages, think about
whether you can trust the other player's message, and make your own decision.

Prompt 2: Please carefully analyze the negotiation messages and make your own
decision.

Prompt 3: Carefully analyze and think about whether you can trust the other
player's message, and then make your own decision.

Prompt 4: You can choose your own choice regardless what the other player
says.

Prompt 5: You should make your own choice regardless what the other player
says.

Prompt 6: You must make your own choice regardless what the other player says.
```

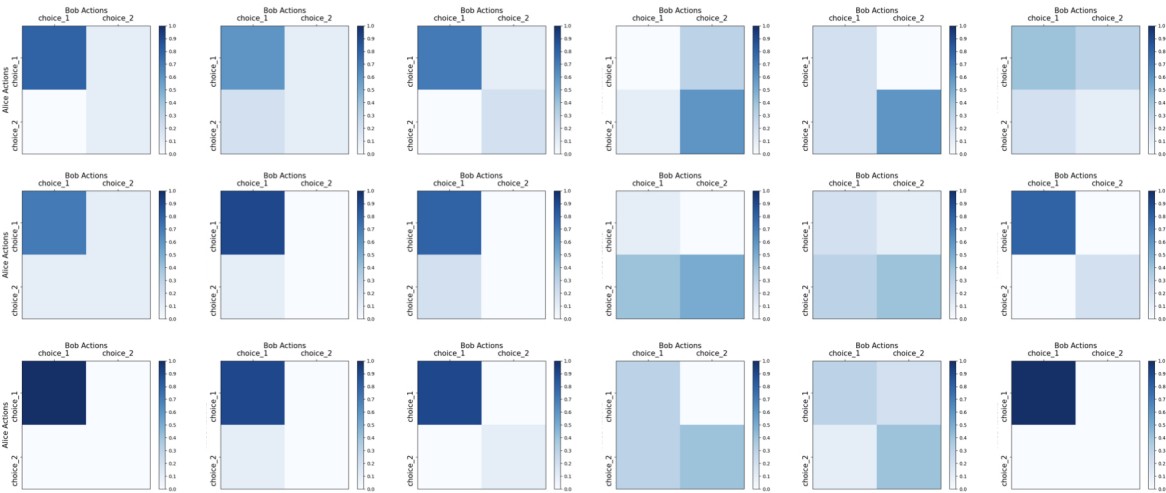

Figure 9: The effect of the 6 engineered prompts on Prisoner's Dilemma game with different rounds of negotiation: 1-round, 2-round, and 3-round for the three rows

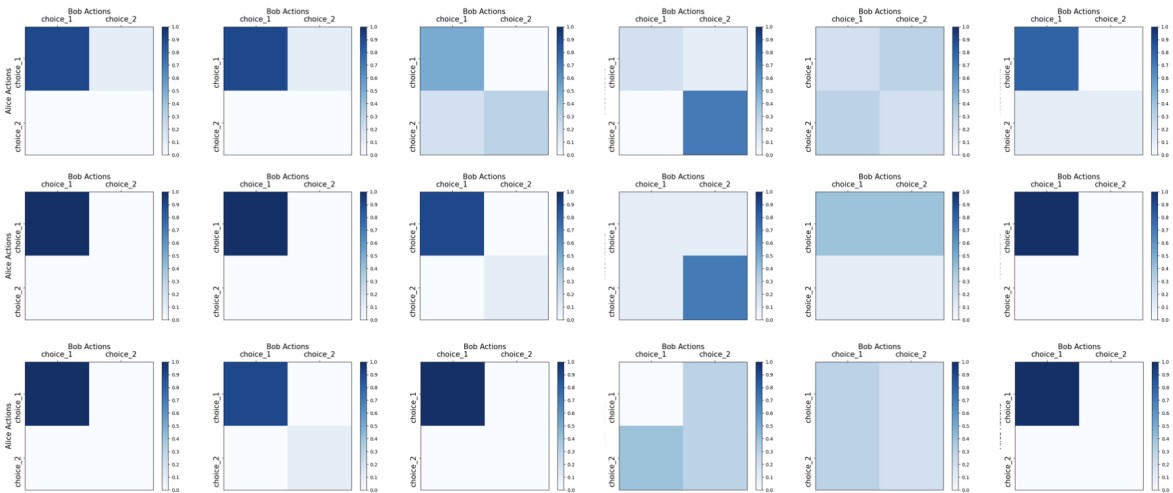

Figure 10: The effect of the 6 engineered prompts on Stag Hunt game with different rounds of negotiation: 1-round, 2-round, and 3-round for the three rows

By comparing the performance of LLM-based agents across these six prompts, we aim to determine whether prompt engineering can effectively control the impact of negotiation on player rationality and improve

decision-making outcomes in various game-theoretic scenarios. Figure 10 and Figure 10 illustrate the impact of the six designed prompts on the outcomes of Prisoner's Dilemma and Stag Hunt games, respectively. In each figure, the six columns correspond to the specific prompts used, while the three rows represent the number of negotiation rounds between the two players before they decide on their actions.

By analyzing these figures, we can assess the effectiveness of each prompt in influencing the players' decision-making processes and evaluate whether prompt engineering can mitigate the effects of negotiation on player's choices as the number of negotiation round increases.

In observation, we can see the following situations:

1. Prompt 1, 2, 3, and 4 do not significantly impact the distribution of strategies chosen by the LLM-based agents in both Prisoner's Dilemma and Stag Hunt games. Even when these prompts explicitly ask the agents to consider the trustworthiness of the other player, they do not lead to a more rational strategy selection.

2. The prompts have varying degrees of influence on the players' decisions, with Prompt 5 and 6 exerting the most significant impact. In the Prisoner's Dilemma, these prompts completely alter the distribution from a heavy focus on the dominated strategy (action 1, action 1) to the dominant strategy (action 2, action 2). In the Stag Hunt, Prompt 5 and 6 also change the distribution from the pareto optimal strategy (action 1, action 1) to a non-optimal strategy (action 2, action 2) or a mixture of strategies.

3. The impact of the prompts can be gradually diminished as the number of negotiation rounds increases. For instance, in the Prisoner's Dilemma, the distribution shifts more towards the dominated strategy as the number of negotiation rounds grows. Similarly, in the Stag Hunt, the distribution moves more towards the pareto optimal strategy as the number of negotiation rounds increases.

Based on the three observations, we can conclude that the use of engineered prompts does not genuinely enhance the rationality of the LLM-based agents in the Prisoner's Dilemma and Stag Hunt games. The impact of the prompts on the agents' decision-making process follows a similar trend in both games, and their influence is diminished as the number of negotiation rounds increases.

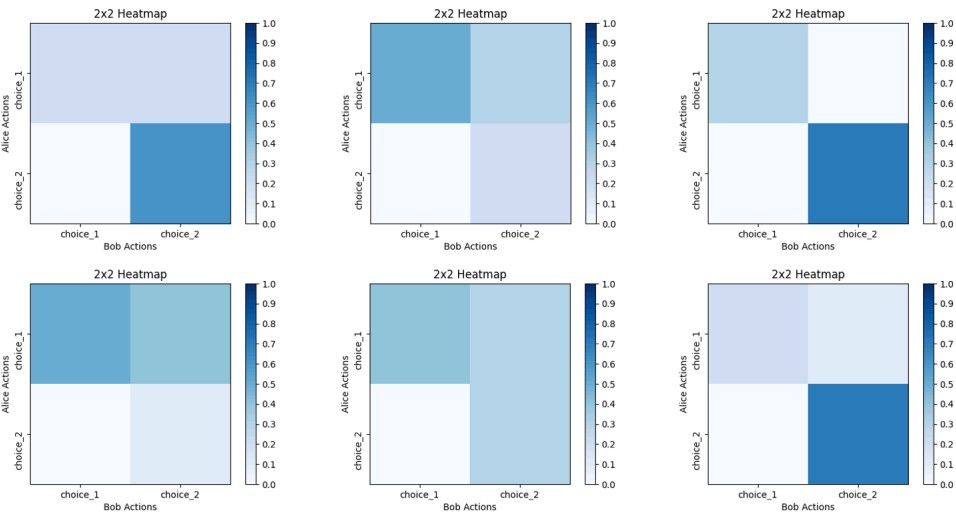

Figure 11: Will the fact that who starts the negotiation affect the result?

### A.9 How does the order of negotiation message affect the action?

It is anticipated that the order in which players initiate negotiations will not influence the final actions taken, particularly in the Battle of the Sexes game. In an ideal scenario, neither the first nor the last player should have a distinct advantage in persuading the other player to act in their favor. To ensure that this is indeed the case, we conduct experiments in the Battle of the Sexes game with varying negotiation rounds (1, 2, and 3) and alter the player who initiates the negotiation, with either player 1 or player 2 starting the negotiation process. This experimental design allows us to investigate the potential impact of negotiation order on the outcomes of the game and assess the fairness of the negotiation process between the two players.

Based on the observation from Figure 11, it is evident that there is no significant change in strategy when the player initiating the negotiation is varied. This observation suggests that the order in which players commence negotiation does not have a substantial impact on the final actions taken in the Battle of the Sexes game.

### A.10 Irrationality Compared with Humans

The previous observations suggest that Large Language Models (LLMs) lack robust rationality across various scenarios. However, it is well-established that humans, too, do not always behave rationally (Dawes & Thaler, 1988; Sally, 1995). This raises an intriguing question: do the irrational tendencies of LLMs mirror those of humans?

To investigate this, we adopt a game setting from a televsion game show called Golden Balls where two contestants play a variant on the classic Prisoner's Dilemma: In this game, two players independently decide to either "split" or "steal" the jackpot. If both choose to split, they share the jackpot equally. If one player chooses to split and the other steals, the stealer takes the entire jackpot. If both players steal, neither receives any money. Human performance of the game is collected by (Van den Assem et al., 2012), where players chose to cooperate (i.e., split the jackpot) 53% of the time on average, a figure consistent with earlier laboratory studies (Dawes & Thaler, 1988; Sally, 1995). The decision to cooperate was sensitive to the size of the jackpot, with cooperation rates decreasing as the stakes increased.

We configured the LLMs to play this game, using the same jackpot sizes as in the human data. Each jackpot size was tested 20 times, resulting in 40 decision-making instances for each size. We then calculated the cooperation rates for the LLMs for different jackpot sizes.

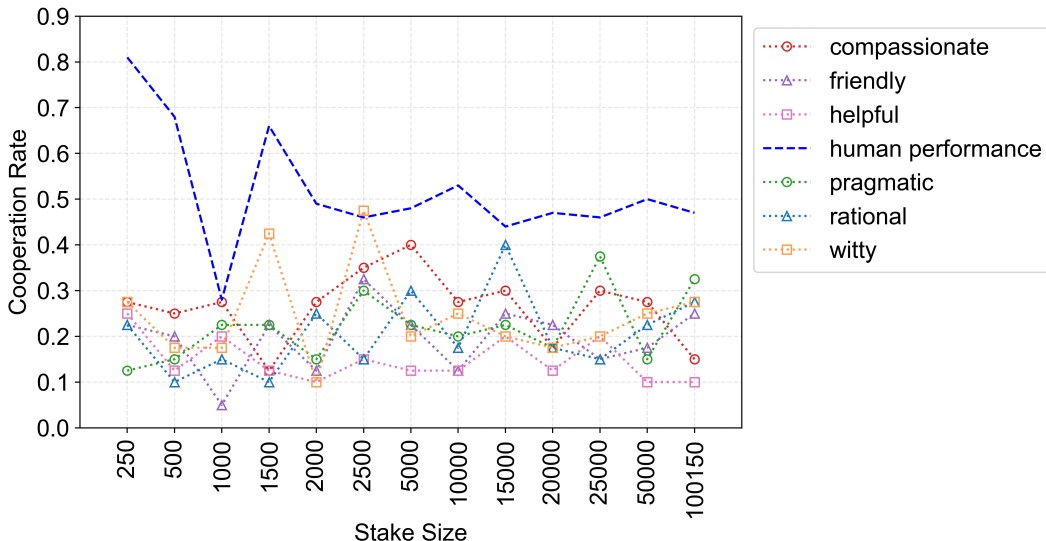

Figure 12: Rationality analysis on Golden Balls game

The results of this comparative analysis are presented in Figure 12. Human cooperation rates were indeed sensitive to the size of the jackpot, with higher cooperation rates for smaller jackpots. This trend is also observed in (Post et al., 2008). In contrast, the LLMs' decision to cooperate was largely insensitive to the jackpot size, regardless of the "personality" prompt used: rational, witty, pragmatic, helpful, friendly, compassionate. Furthermore, the LLMs' cooperation rates were generally lower than those of the humans.

While this analysis provides a preliminary comparison of irrational tendencies in LLMs and humans, it is by no means exhaustive. Further research is needed to establish a more comprehensive understanding of the relationship between human and LLM irrationality.

---

**Summary for raw-LLM v. workflow-LLM and workflow-LLM v. raw-LLM**

- **Lack of Robustness to Numerical Variations**: Empirical results indicate that LLMs either do not exhibit rationality or their rationality is highly sensitive to numerical changes. When the payoff matrix undergoes perturbations that leave the Nash Equilibrium unchanged, the performance of LLMs varies significantly.

- **Impact of Negotiation on Rationality**: Consistent with observations made in Section A.3, we find that rational choices are undermined by the introduction of negotiation, even in the absence of grounds or guarantees for trust.

- **Effect of Prompt Variation on Negotiation Impact**: The wording of the prompt can mitigate the influence of negotiation on rationality; however, this mitigating effect diminishes as the number of negotiation rounds increases.

- **Order of Negotiation Initiation**: The sequence in which players initiate negotiation does not significantly affect the game's outcome, even in coordination games such as the Battle of the Sexes.

- **Comparison of Irrationality Between LLMs and Humans**: Although LLMs lack robust rationality across various scenarios, the nature of their irrationality differs from that observed in human decision-making.

---

