# OpenReview forum: "Game-Theoretic LLM: Agent Workflow for Negotiation Games"
_TMLR — Rejected by TMLR_

### Review · Reviewer_8BLc · 2025-05-13

**Summary Of Contributions:**

The paper explores how LLMs make strategic decisions in various complete and incomplete-information games. Decisions were evaluated in terms of whether they were playing an NE or not. The work attempted to establish some baselines and then introduces several "workflows" to help coax LLMs into making decisions more rationally. It then examines a meta decision process around which workflows to deploy and when to maximize rational decision making in LLMs. The paper also motivates why strategic decision making is important for LLMs.

**Audience:**

Yes

**Broader Impact Concerns:**

No ethical concerns.

**Claims And Evidence:**

No

**Requested Changes:**

Please address and answer the questions I have stated above. Later sections did get into the details, but I was struggling to connect them to the baselines with clarity. I believe the paper needs to be rewritten more tersely and key details not omitted.

Flow:
1. There are details in Section 4.4.1 which I think are necessary to understand Section 4.3.

Spelling/Grammar:
1. "Nash Equilibria" -> "Nash equilibria"
2. "pareto" -> "Pareto"

Line-by-line:
1. [Page 2] "Cringed of these well-defined..."
2. [Page 4] "corresponds on the"
3. [Page 5] Section 4.3, additional parenthesis
4. [Page 14] "propose proposes"

**Strengths And Weaknesses:**

Strengths:

1. Strategic interactions between LLMs is very worthy of study.
2. Figures were well designed.

Weakness:

1. The paper is both longer than it needs to be. The writing should be made more succinct.
2. While also lacking crucial details in frustrating places. For example, in Section 4.3 I was immediately confused. Was Table 1 compiled in a single-agent fashion by just querying an LLM how it would play in a game? But if so, how are you checking for selecting an NE in a game like Battle of the Sexes which requires coordination? Does the LLM suggest the full joint strategy (A,A) or (B,B)? Also I guess you are just looking at pure NEs? If multiple LLMs are used here, are they playing with each other (different models)? Similarly for Table 2 stating that there are "4 rounds of negotiation" is not enough information to work out exactly what the table is showing. Are they negotiating with themselves? Against another agent, but same model? Do they observe each other's chain of thought? Does one get to act and the other observe and then act themselves. Negotiation is sequential, but these games are simultaneous action.
3. Normal-form games cease to be normal-form games when multiple rounds and negotiation is added. They become extensive-form games with different equilibrium concepts and different equilibria. I don't think this nuance was discussed in the paper.
4. NE is a prescriptive  solution concept in two-player zero-sum games. In other games it is not clear what the best solution concept is. I think NE == rational should be stated as an assumption in this work. To be clear, I think this is the right assumption to make for this study, but it should be stated as an assumption.

Questions:

1. [Section 1]  "This is the first attempt to systematically integrate..." You have cited a bunch of that seem to integrate game-theory and LLMs. Is the word"classic" doing heavy-lifting? I am not sure if "classic game-theoretic strategies" is well defined.
2. [Section 4.3] How are the LLMs prompted with the game description?
3. [Section 4.3] Are you only considering pure NEs?
4. [Section 4.3] These are multiagent games. Are the agents playing with themselves?
5. [Section 4.3] Has the negotiating mechanism been described?
6. [Section 4.3] For simultaneous games, how do the agents negotiate?
7. [Section 4.3] The text says "out of 20", but the text before and Tables imply "out of 10"?

---

> ### Author Response · Authors · 2025-06-07
> **Answer to Questions 1- 4**
>
> > Question 1.1: This is the first attempt to systematically integrate..." You have cited a bunch of that seem to integrate game-theory and LLMs.
>
> Thank you for this important clarification regarding our novelty claims.
>
> While there has indeed been substantial research at the intersection of game theory and LLMs, we argue that our work represents **the first systematic attempt to enhance LLM strategic decision-making through embedded game-theoretic algorithms**, rather than merely evaluating LLM behavior in game-theoretic settings or using game theory as an external steering mechanism.
>
> To clarify the distinction from existing work:
>
> **Prior work focuses primarily on evaluation:** Most existing research, as we discuss in our related work section (CogBench; GTBench; Jen-tse Huang et al. 2024; Fulin Guo. 2023; Elif Akata et al. 2023; Nunzio Lorè and Babak Heydari 2023), evaluates LLM rationality, coordination abilities, or risk tolerance in game-theoretic scenarios without fundamentally improving their strategic reasoning capabilities.
>
> **The most relevant prior work**: "Steering Language Models with Game-Theoretic Solvers" (Gemp et al.), fundamentally differs from our approach in that they use external game-theoretic solvers (CFR) to compute optimal strategies and then steer LLM responses through prompting; whereas we embed the algorithmic reasoning process directly into the LLM's reasoning workflow, teaching LLMs to reason strategically rather than providing them with external strategic guidance.
>
> > Question 1.2: Is the word"classic" doing heavy-lifting? I am not sure if "classic game-theoretic strategies" is well defined.
>
> Thank you for pointing out this imprecision in our language. You are correct that "classic game-theoretic strategies" is not well-defined terminology. What we mean to convey is that for **complete-information games**, we utilize fundamental, well-established game-theoretic algorithms - specifically **dominant strategy analysis**, **backward induction**, **Bayesian belief updating**, can be found in textbooks such as "Game Theory, Interactive Strategies in Economics and Management" published by Cambridge University Press and "Game Theory 101: The Complete Textbook" by William Spaniel.
>
> > Question 2: How are the LLMs prompted with the game description?
>
> In complete information game, each LLM agent receives a structured prompt that includes: (1) a clear description of the game rules and objective, (2) the complete payoff matrix showing all possible outcomes for both players, (3) their role designation (Player 1 or Player 2), and (4) the available actions they can choose from. For example, in the Prisoner's Dilemma, the prompt explicitly states the scenario (two suspects being interrogated), defines the actions (Cooperate/Defect), and presents the payoff matrix with clear numerical outcomes for each action combination. The agents are instructed to analyze the strategic situation and select their preferred action. Details can be found in the anonymized github's python files: src/simultaneous_game_agent.py (for simultaneous game setting) src/sequential_game_agent.py (for sequential game setting), and workflow_design (for the incomplete-information game).
>
> In in-complete information games, each player will only be provided private information of itself. Otherwise, it is the same as complete-information games.
>
> > Question 3: Are you only considering pure NEs?
>
> Yes, we focus exclusively on pure strategy Nash Equilibria in our analysis. **We deliberately excluded mixed-strategy Nash Equilibria due to the prohibitive experimental costs involved in their evaluation.**
>
> Analyzing mixed-strategy equilibria with LLMs would require a multi-layered sampling approach: (1) **Strategy identification**: We would need numerous samples from each LLM to infer what mixed strategy it is employing, (2) **Strategy stability assessment**: We would then need many additional experimental runs to determine whether these inferred mixed strategies remain stable across different sessions or if the mixing probabilities themselves vary. Therefore extending to mixed-strategy analysis would increase our experimental computation a lot.
>
> In our initial explorations, we included mixed-strategy games such as Rock-Paper-Scissors, but we removed them due to the extremely high computational cost of obtaining meaningful results across different experiment settings.
>
> > Question 4: These are multiagent games. Are the agents playing with themselves?
>
> No, the agents are not playing with themselves. **We use a multi-agent setup where two separate LLM instances interact.** These could be either two instances of the same model (e.g., two Claude-3.5 Sonnet agents) or two different models (e.g., Claude-3.5 Sonnet vs. GPT-4o). Each agent independently processes the game information and makes decisions without access to the other agent's internal reasoning process or private information (if any).

---

> ### Author Response · Authors · 2025-06-07
> **Answer to Question 5 - 7**
>
> > Question 5: Has the negotiating mechanism been described?
>
> We employ two distinct negotiation mechanisms depending on the information structure of the game:
>
> **For Complete-Information Games (Section 4):** The negotiation mechanism consists of an unstructured, free-form discussion format where agents exchange natural language messages without predetermined conversational constraints. This is becuase in our experimental results (Tables 2 and 4) for complete information games, we noticed that the primary challenge for LLMs in complete-information games is not coordination through communication, but rather making rational strategic choices in the first place. Thus we don't think developing sophisticated communication workflows would be necessary.
>
> **Thus for complete-information game, there is no specific negotiating mechanism to be described.**
>
> **For Incomplete-Information Games (Section 5):** The negotiation mechanism follows the workflow that is integral to our algorithmic contribution. Unlike the free-form communication in complete-information games, negotiation here serves as an active information inference process where agents systematically analyze opponents' proposals and statements to deduce private valuations through Bayesian updating.
>
> This structured negotiation method is formally described in Section 5.2, including: (1) algorithmic frameworks for proposal generation and evaluation, (2) Bayesian belief updating mechanisms based on observed negotiation behavior, and (3) strategic reasoning processes that gradually transform incomplete information into actionable complete information. The negotiation mechanism is not merely communicative but constitutes a core component of our novel algorithmic approach to incomplete-information games.
>
> **Thus for incomplete-information game, the main focus is the negotiation workflow and it is carefully described in Section 5.2, especially in "Allocation Proposal" part.**
>
> > Question 6: For simultaneous games, how do the agents negotiate?
>
> For simultaneous games, "negotiation" occurs as a pre-play communication phase in a free-communication form before the actual game actions are taken.
>
> The process involves:
>
> (1) **Free discussion phase**: agents engage in open conversation for multiple rounds, sharing whatever information they deem relevant, discussing potential strategies, or attempting coordination without any prescribed format,
>
> (2) **Action commitment**: after the discussion concludes, both agents independently and simultaneously select their actions without further communication,
>
> (3) **Outcome determination**: the game outcome is determined based on the simultaneous action choices.
>
> Our primary interest is in observing whether this free-form sharing of intentions or information actually changes the agents' minds regarding their final actions, particularly comparing coordination games (where information sharing should matter) with non-coordination games like Prisoner's Dilemma (where it theoretically shouldn't affect rational play).
>
> > Question 7: The text says "out of 20", but the text before and Tables imply "out of 10"?
>
> Thank you for catching this potential confusion in our description.
>
> To clarify: for each game configuration, we conduct 10 independent experimental runs. Since each game involves 2 agents (players) and each agent makes one decision per game, this results in a total of 20 individual decisions across the 10 runs **(10 runs × 2 agents per run = 20 decisions).**
>
> When we report results such as "Claude-3.5 Sonnet selects the Pareto optimal strategy in 5 out of 20 total choices," we are referring to the total number of individual agent decisions across all runs, not the number of game instances. Similarly, when we mention "11 out of 20" after negotiation, this represents the count of individual agent choices that resulted in the Pareto optimal strategy.

---

> ### Author Response · Authors · 2025-06-07
> **Other comments in Requested Changes and Weakness**
>
> > Grammar Errors: thank you very much for the careful reading. We will address all the spelling, grammar, and typographical errors you identified:
>
> "Nash Equilibria" → "Nash equilibria" (throughout)
> "pareto" → "Pareto" (throughout)
> "Cringed on these well-defined..." → "Grounded in these well-defined..."
> "corresponds on the" → "corresponds to the"
> Remove additional parenthesis in Section 4.3
> "propose proposes" → "proposes"
>
> > Omitted Details
>
> To make our experimental methodology crystal clear, we will add comprehensive examples in the appendix, including:
>
> **Complete example prompts:** Full prompts showing exactly how we present games to LLM agents, including game descriptions, payoff matrices, and instruction formats
>
> **End-to-end interaction examples:** Complete transcripts of actual game runs showing the full sequence from initial prompts through negotiation (if applicable) to final action selection
>
> **Step-by-step workflow demonstrations:** Detailed examples of how our workflows guide LLM reasoning in both complete and incomplete information settings
>
> This should eliminate any ambiguity about how our games are conducted and allow readers to fully understand our experimental procedures.
>
> > Weakness 1
>
> I think most questions raised in the first point of weakness are also presented in the Questions part.
>
> > Weakness 2: Normal-form games cease to be normal-form games when multiple rounds and negotiation is added.
>
> Thank you for this important theoretical observation! This is indeed a crucial point that we should discuss explicitly.
>
> We want to clarify that in our experiments for complete-information games, we implement "negotiation" as **cheap talk**. Specifically, it
>
> (1) preserve simultaneous structure: While agents engage in pre-play communication, the actual game actions remain simultaneous and non-observable to opponents;
>
> (2) non-binding communication: Negotiation messages carry no commitment power - agents can discuss any strategy but still choose actions independently;
>
> (3) no direct payoff consequences: The communication itself does not affect payoffs - only the final simultaneous actions determine outcomes;
>
> Adding this free communication in complete-information games in our paper **allows us to study how communication affects the behavior of LLM-based agents**. It is a standard methodology in experimental game theory for examining the role of pre-play communication in coordination and strategic behavior.
>
> We will explicitly discuss this game-theoretic distinction in our revision. Thank you for highlighting this important theoretical nuance.
>
> > Weakness 3: I think NE == rational should be stated as an assumption in this work.
>
> Thank you very much for pointing this out. We will definitely mention and potentially discuss it in the revised version.

---

### Review · Reviewer_ya8G · 2025-05-19

**Summary Of Contributions:**

The paper studies the intersection of game theory and LLMs, specifically whether modern LLMs make rational strategic choices and if they can be improved through designed workflows. The authors evaluate this in three tests: 1) how closely do LLMs follow Nash equilibrium play in both complete and incomplete information games. 2) Does embedding game theoretic reasoning steps through prompting improve performance? 3) Given heterogeneous opponents, is it rational to adopt a workflow? The results on multiple games indicate that workflows improve NE and fairness metrics in most settings.

**Audience:**

Yes

**Broader Impact Concerns:**

More rational LLM negotiators could be weaponised for adversarial uses (like automated price-haggling exploitable to defraud inexperienced users). A brief ethics discussion or even a single sentence in the conclusion would signal awareness of potential misuse.

**Claims And Evidence:**

No

**Requested Changes:**

- Most important here are the statistical significance tests (or bootstrapped confidence intervals for NE and fairness). Note that this is the reason I am marking "No" on the clear evidence question below. It should be fairly easy to add in clear confidence intervals or p-values for significance tests.
- Better discussion on how the temperature of o1 could be affecting the results.
- Ablation experiments for workflow components. What is the importance of each component? This is a "nice to have", but not required.

**Strengths And Weaknesses:**

### Strengths
- Well motivated research that tackles an interesting and well-defined problem in game theoretic reasoning. It ties theoretical guarantees (Nash equilibrium) with practical applications (negotiation agents)
- Multiple factors are evaluated including rationality, fairness and meta-strategy to give a more complete picture of LLM behaviour
- The paper is well written overall with clear diagrams making it easier to follow
- The results are interesting and will be of value to both the ML and game theory research communities

### Weaknesses
- The experiments don’t include ablation of workflow components. This makes it hard to gauge what the main drivers of performance gains are and how much is coming from dominant strategy search vs CoT invocation
- Although the relevant works section is good overall, it might be worth looking at irrational moves with respect to humans as discussed in Camerer (2003) Behavioral Game Theory: Experiments in Strategic Interaction. This might provide a deeper viewpoint into LLM choices and if they reflect or diverge from humans
- The payoff matrices are normalised to [0, 1] (Section 2), which could compress incentive differentials and exaggerate apparent gains. Also, all games are two-player, zero-sum or constant-sum. Extending to variable-sum or stochastic games might reveal new insights or biases in the workflow efficacy
- The statistical power is often limited, 10 repeated runs on only 50 hardest dialogues. Variances/errors are not always shown making it difficult to gauge whether differences are significant

---

> ### Author Response · Authors · 2025-06-09
> **Reply to Weakness 1, 2, 3**
>
> > Weakness 1: ablation study
>
> Thank you for this suggestion. We acknowledge that traditional component-wise ablation studies would be valuable, though our workflow designs present some inherent challenges for standard ablation approaches.
>
> **For Complete-Information Games:** Our workflow integrates two fundamental game-theoretic algorithms: dominant strategy analysis and backward induction. These represent well-established, atomic algorithmic components that are difficult to meaningfully decompose further. However, we do provide implicit ablation through our comparison between workflow and non-workflow conditions (Tables 5-6 vs. Tables 7-8), which demonstrates the overall effectiveness of incorporating these algorithmic reasoning steps.
>
> **For Incomplete-Information Games:** Our workflow centers on Bayesian belief updating to infer opponents' private valuations, gradually transforming incomplete-information games into complete-information settings. While the Bayesian updating process itself is a unified algorithm, we provide extensive analysis of its key component—**inference accuracy**—which is the critical factor determining workflow success.
>
> Specifically, we present detailed evaluation of valuation estimation performance in Tables 14-16 (pages 27-29), showing:
>
> - **Precision**: 79.5%-95.5% across models (Table 14)
>
> - **Recall**: 27.4%-37.7% across models (Table 14)
>
> - **Temporal dynamics**: How estimation accuracy evolves across negotiation rounds (Table 15)
>
> - **Indistinguishable set analysis**: Performance with respect to strategically equivalent valuations (Table 16)
>
> This analysis effectively serves as an ablation study of the inference component, demonstrating that workflow success correlates strongly with accurate belief updating.
>
> > Weakness 2
>
> Thank you for pointing this out and for the excellent reference! We are indeed very interested in behavioral economics and the rich literature on human deviations from rational play and will add relevant discussion in Related Work section.
>
> But just to mention, **in appendix, we do touch on the comparison between LLM and human irrationality in Appendix A.10 on page 43**, where we analyze LLM behavior in the **Golden Balls game** (a Prisoner's Dilemma variant) and compare it to human cooperation patterns documented in the behavioral economics literature. Interestingly, we find that while both LLMs and humans deviate from pure rationality, their patterns of irrationality differ significantly. For instance, humans show sensitivity to stake size in cooperation decisions, whereas LLMs demonstrate relative insensitivity to payoff magnitudes.
>
> We agree that a more comprehensive analysis drawing on Camerer's framework and other behavioral game theory findings would be valuable. We have **ongoing work that explores non-rational factors for LLM decision-making**, where the rational behavior we study here functions as a baseline for understanding systematic deviations. This could include examining and steering LLM behavior with risk aversion, loss aversion, anchoring biases, and other well-documented human behavioral patterns.
>
> > Weakness 3: The payoff matrices are normalised to (Section 2)
>
> Thank you for this comment, but I believe there may be a misunderstanding.
>
> **We did not normalize the payoff matrices in our experiments**. The payoff matrices we used are presented exactly as shown in Appendix A.2, without any normalization procedures applied. For the complete-information games (Section 4), we use the standard payoff matrices from the game theory literature without any modifications.
>
> Additionally, **we specifically study the impact of numerical values by systematically perturbing payoff matrices** while preserving the Nash equilibria structure (Section 6.2 and Appendix A.5). This analysis demonstrates that LLMs are indeed sensitive to numerical variations even when the strategic structure remains unchanged, which would not be possible if we had normalized the matrices.
>
> Could you clarify which specific payoff matrices or normalization procedure you are referring to? We want to ensure there is no misunderstanding about our experimental methodology.
>
> (reply to other comments to be continued)

---

> ### Author Response · Authors · 2025-06-10
> **Reply to Weakness 4 and Requested Changes 2**
>
> > Weakness 4: missing statistical significance tests
>
> Thank you for raising this important concern about statistical rigor. We acknowledge that computational constraints limited our sample size, but we have conducted proper statistical significance testing on our results.
>
> **Experimental Constraints:** Each experimental run is computationally expensive, requiring multiple LLM API calls per negotiation round across extended dialogues. To balance statistical rigor with computational feasibility, we focused on the most challenging datapoints.
>
> Statistical Analysis with Standard Deviations:
> | Model | Negotiation Round | Agreement | Alice score | Bob score | PO | EF | total reward |
> | ------ | ----------------- | --------------- | ------------ | ------------ | ---------------- | ---------------- | ------------- |
> | Sonnet | 6.97 +-1.31 | 0.9409 +-0.0265 | 5.49 +- 0.37 | 5.57 +- 0.24 | 0.6863 +- 0.0464 | 0.6818 +- 0.3033 | 11.06 +- 0.50 |
> | o1 | 4.21 +-1.87 | 0.7820 +- 0.043 | 4.11 +- 0.29 | 4.18 +- 0.41 | 0.4818 +- 0.0526 | 0.4773 +- 0.0452 | 8.29 +- 0.57 |
> | GPT-4o | 18.81 +-0.75 | 0.7864+-0.0568 | 2.70 +- 0.29 | 4.34+-0.27 | 0.3691 +- 0.0526 | 0.3409 +- 0.5183 | 7.14 +- 0.45 |
> | Opus | 4.41 +-2.02 | 0.4636+- 0.0369 | 2.64 +- 0.33 | 2.87+- 0.39 | 0.3545 +- 0.0636 | 0.2591 +- 0.0432 | 5.51 +- 0.58 |
>
> **Statistical Significance Testing:** We performed paired t-tests comparing raw-LLM vs. workflow-LLM performance for each model. The p-values demonstrate highly significant improvements:
>
> **Workflow-LLM vs. Workflow-LLM Results (Mean ± SD):**
> | Model | Negotiation Round | Agreement | Alice score | Bob score | PO | EF | total reward |
> | ------ | ----------------- | ------------ | ------------ | ------------ | --------------- | --------------- | ------------- |
> | Sonnet | 4.38 +- 0.88 | 1.00 +- 0.00 | 5.92 +- 0.04 | 6.23 +- 0.04 | 0.7773 +- 0.027 | 0.9924 +- 0.011 | 12.15 +- 0.07 |
> | GPT-4o | 5.32 +- 1.28 | 1.00 +- 0.00 | 5.87 +- 0.09 | 6.34 +- 0.08 | 0.8591 +- 0.017 | 0.9848 +- 0.011 | 12.21 +- 0.16 |
> | Opus | 4.11 +-0.73 | 1.00 +- 0.00 | 5.95 +- 0.06 | 6.47 +- 0.05 | 0.8727 +- 0.01 | 0.9545 +- 0.019 | 12.22 +- 0.10 |
>
> **Statistical Significance Testing:** We performed paired t-tests comparing raw-LLM vs. workflow-LLM performance for each model (n=10 runs each). The p-values demonstrate highly significant improvements:
>
> | Model | Negotiation Round | Agreement | Alice score | Bob score | PO | EF | total reward |
> | ------ | -------------------- | --------- | ----------- | --------- | ---------- | --------- | ------------ |
> | Sonnet | 0.0063 | 0.0011 | 0.032 | 0.0003 | 0.0053 | 0.0514 | 0.0013 |
> | GPT-4o | 3.5783e-08 | 3.045e-05 | 1.190e-06 | 2.844e-06 | 4.372e-08 | 0.0240 | 1.0561e-08 |
> | Opus | 0.7627 | 8.747e-10 | 1.879e-08 | 3.390e-08 | 9.3173e-08 | 7.853e-10 | 6.008e-09 |
>
> **The consistently low p-values across metrics and models provide strong evidence for workflow effectiveness**. We acknowledge that larger sample sizes would strengthen our conclusions and will pursue more extensive evaluation in future work as computational resources permit.
>
>
> > Requested Changes 2: impact of temperature on o1
>
> Thank you for this suggestion. We want to clarify that the temperature parameter is actually **not supported with the o1 model**: it uses a fixed, default temperature setting that cannot be adjusted by users.
>
> However, **we do provide analysis of temperature effects using GPT-4o in Section 5.5.1 (page 17, Table 7)** for the incomplete-information games, where we demonstrate that the impact of temperature when the LLM has no workflow assisted: higher temperature (1.0) leads to better agreement rates, total rewards, and envy freeness compared to lower temperature (0.0), likely due to increased exploration of strategic alternatives. When the LLM has no workflow assisted, we also examine temperature effects with GPT-4o in Section 5.5.2 (page 19, Table 9), finding that workflow-guided agents exhibit much greater stability across temperature settings - both temperature 0.0 and 1.0 achieve 100% agreement rates and similar performance metrics. This suggests that our workflows help stabilize LLM strategic behavior regardless of temperature-induced variability.

---

> ### Author Response · Authors · 2025-06-12
> **Reply to Broader Impact**
>
> Thank you for raising this very important consideration regarding potential misuse. We agree that the societal implications of having more strategically capable LLM agents.
>
> While our work does raise concerns about potential adversarial applications such as automated systems that could exploit inexperienced negotiators in commercial settings, we also believe the technology has significant potential for positive societal impact. Enhanced strategic reasoning capabilities could serve as **equalizing tools** that help less experienced or disadvantaged users navigate complex negotiations more effectively.
>
> For example, strategically enhanced LLMs could assist consumers in fair price negotiations, help individuals navigate complex contract discussions, or provide strategic guidance to users who lack negotiation experience. In this sense, **the technology could democratize access to strategic reasoning capabilities rather than concentrate advantage among already sophisticated actors.**
>
> However, we fully acknowledge that such capabilities should be developed and deployed with caution. The potential for misuse requires careful consideration of regulation and safeguards to prevent exploitation of vulnerable populations. Future work should explicitly address these ethical considerations and include mechanisms to ensure responsible deployment.

---

### Review · Reviewer_AoFU · 2025-06-03

**Summary Of Contributions:**

The paper begins by examining the ability of several state-of-the-art LLMs to make strategic decisions in multi-player settings. It finds that many common LLMs fail to make rational decisions even in relatively simple settings. The paper then proposes a game-theoretic pipeline intended to augment the game-theoretic reasoning of LLMs. Experiments show that this pipeline leads to a significant improvement in identifying optimal strategies.

**Audience:**

Yes

**Broader Impact Concerns:**

No concerns about the ethical implications of the work.

**Claims And Evidence:**

Yes

**Requested Changes:**

As I explained above, the main critique I have of the paper is that it fails to justify its novelty and significance in the presence of the existing body of work. Most of the questions studied have been thoroughly addressed in recent papers. Can the authors provide more details about the novelty of the results? How do the results of this paper differ from existing ones?

A minor comment:
- footnotes should come after punctuation marks

**Strengths And Weaknesses:**

On the positive side, the paper addresses an interesting and natural question. Understanding and improving the strategic capabilities of LLMs in multiagent settings is a central question going forward, and has already received a lot of attention in recent years, as discussed by the authors. So the motivation of the paper is certainly concrete. The writing and structure of the paper are also generally of high quality, and the paper clearly conveys the key ideas and findings.

That being said, at the present state of the manuscript, I have serious concerns about the novelty and the significance of the results. In the last few years, there have been many papers that pursue this very direction--many of which are cited in this paper. In terms of evaluating
the game-theoretic capabilities of common LLMs, the findings of this paper mirror prior results. Where exactly is the novelty? What differentiates this paper from earlier work? I understand that there are certain variations in terms of the LLMs studied and the type of games tested on, but that doesn't seem to go far enough in terms of a novel contribution. While the paper provides many references, it fails to provide a meaningful comparison between the results. The same critique holds for the second part of the paper that investigates how to improve the strategic capabilities of LLMs. The paper cites recent papers in the same direction, such as Gemp et al. (2024), but does not explain how the proposed approach is novel and how it improves upon existing approaches. It is not enough to just cite existing papers, the paper must explain how the results differ. That being said, some of the considerations of this paper appear to be new. For example, the analysis of whether the proposed workflow should be adopted or not is interesting and I haven't seen this question been addressed before.

---

> ### Author Response · Authors · 2025-06-07
> **Reply to Weakness and Requested Changes about paper novelty**
>
> Thank you for your thoughtful review and for acknowledging the importance of understanding LLM strategic capabilities in multiagent settings. We appreciate your feedback regarding novelty and significance. Here is our clarification and detailed comparison with previous work:
>
> **Addressing Novelty Concerns:**
>
> We understand your concern about distinguishing our work from the existing literature. While there has indeed been substantial research in this area, we believe there is an important distinction between **evaluation work** and **enhancement work** that clarifies our contributions:
>
> **Prior Work Context:** Most existing research at the intersection of game theory and LLMs focuses primarily on **evaluation**, e.g. assessing LLM rationality, coordination abilities, risk tolerance, and strategic behavior across various game-theoretic scenarios, as stated in Section 2. These studies have established that LLMs exhibit strategic limitations, but they do not fundamentally improve LLM strategic reasoning capabilities. Our work addresses the next crucial step: **enhancement** of LLM strategic decision-making through algorithmic integration over a diverse landscape of game-theoretic scenarios.
>
> **Our Distinct Contributions:**
>
> **1. Embedded Algorithmic Reasoning vs. External Steering:** The most relevant comparison is with Gemp et al. (2024), which represents a fundamentally different approach. Their work uses external game-theoretic solvers (CFR) to compute optimal strategies and then steers LLM responses through prompting with computed solutions. Crucially, they model natural language as discrete actions (7 days in scheduling meetings, 4 tones in trading fruit, 3 argument types in debate) and apply existing solvers to these predefined action spaces.
>
> In contrast, our approach embeds game-theoretic algorithmic reasoning directly into the LLM's internal decision-making process. For incomplete-information games, we do not discretize natural language responses into categorical types, which would lose substantial information. Instead, we focus on algorithmically deriving private information from the rich content of opponents' natural language responses to enable better decision-making. While our Deal-or-No-Deal setting is similar to their trading fruit scenario, we develop our own novel algorithm based on Bayesian updating that utilizes the full expressivity of natural language rather than reducing it to a small set of categories of actions.
>
> In both complete-information games and incomplete-information games, we leverage workflows to teach LLM how to think and reason and then make strategic moves, instead of utilizing a solver for a solution and then to steer the LLM.  **When LLMs perform the reasoning steps themselves, we believe there are several advantages:**
>
> (1) they can integrate strategic analysis with their natural language understanding capabilities
>
> (2) adapt their strategic reasoning based on conversational context.
>
> (3) we can observe their strategic thinking process through chain-of-thought so that we can see where strategic reasoning fails or succeeds
>
> (4) potentially, we don't need to split natural language responses into a discrete action space
>
> **2. Novel Algorithmic Contributions for Incomplete Information:** For incomplete-information games, we develop a novel algorithm based on common sense human negotiation principles and Bayesian update for private information inference. Unlike existing approaches that model language as discrete actions (as in Gemp et al.'s framework with predefined action spaces), we focus on algorithmically inferring private information through natural conversational content. Our Bayesian updating approach treats negotiation as a dynamic information revelation process, gradually transforming incomplete information games into complete information settings where optimal solutions become tractable.
>
> **3. Meta-Strategic Analysis:** As you noted, our analysis of workflow adoption decisions represents a novel meta-strategic consideration. We demonstrate that the decision to adopt strategic workflows creates its own game-theoretic problem, with different LLMs exhibiting different Nash equilibria regarding workflow usage. This meta-level analysis opens new research directions in adaptive strategy selection.
>
> ---
>
> **Short Summary:**
>
> - **Methodological**: external steering vs. workflow-aided reasoning
> - **Algorithmic**: Novel Bayesian-inference-based algorithm for incomplete information game
> - **New Direction**: Meta-strategic analysis of workflow adoption
>
> We will enhance our related work section to provide more explicit comparisons with existing approaches, particularly highlighting the distinction between evaluation studies and enhancement methodologies, and clarifying these methodological and algorithmic differences.
>
> Thank you for the formatting note regarding footnotes. We will correct this in our revision.

---

### Comment · Action_Editor_E5iz · 2025-03-12
**From the action editor: paper is very long.**

The paper is abnormally long (42 pages of text), with no appendices. This is likely to cause review delays.  A lot of the paper is generic background from game theory.

I am requesting that you consider either reducing the paper length, putting the background material in the appendix and focussing on the novel contribution or else provide a justification why you really need all those pages.

---

> ### Author Response · Authors · 2025-03-13
> **Reply to action editor**
>
> Thank you for letting me know and I will re-organize the paper to a suitable length.

---

> > ### Comment · Action_Editor_E5iz · 2025-03-26
> > **From the action editor: deadline.**
> >
> > Please note that the date for the decision on whether to progress the paper to the review phase was 18 March.
> > Hence, I am requesting that you update the paper soon.

---

> > > ### Author Response · Authors · 2025-03-27
> > > **Will submit this weekend**
> > >
> > > Hello, thank you for the reminder and I will submit it this weekend.

---

### Decision · Action_Editor_E5iz · 2025-08-23

**Recommendation:** Reject

**Additional Comments:**

I encourage the authors to resubmit the paper as a major revision.

If you do so, please redo all the experiments, perform additional ablations and make sure *all* results are backed by sound statistics.

**Audience:**

Yes

**Audience Explanation:**

Strategic decision making with LLM is a timely and important topic of interest to many in the TMLR audience.

**Claims And Evidence:**

No

**Claims Explanation:**

The paper has issues with evaluation:
- reviewer ya8G has suggested that the paper does not include the necessary ablations
- the tables in the original paper did not include any measure of uncertainty in the outcome (confidence bar or similar). While this was rectified to some extent during the rebuttal, either this should be done for all experiments in the paper or else there should be a clear explanation why this is not necessary.

Overall, I am not convinced by the evaluation methodology as it currently stands.

**Resubmission Of Major Revision:**

The authors may consider submitting a major revision at a later time.